# You Only Sample Once: Taming One-Step Text-to-Image Synthesis by Self-Cooperative Diffusion GANs

**Yihong Luo[1], Xiaolong Chen[2], Xinghua Qu[3], Tianyang Hu[4], Jing Tang[2,1]***

[1] HKUST   [2] HKUST(GZ)   [3] Bytedance Seed   [4] NUS

## Abstract

Recently, some works have tried to combine diffusion and Generative Adversarial Networks (GANs) to alleviate the computational cost of the iterative denoising inference in Diffusion Models (DMs). However, existing works in this line suffer from either training instability and mode collapse or subpar one-step generation learning efficiency. To address these issues, we introduce **YOSO**, a novel generative model designed for rapid, scalable, and high-fidelity one-step image synthesis with high training stability and mode coverage. Specifically, we smooth the adversarial divergence by the denoising generator itself, performing self-cooperative learning. We show that our method can serve as a one-step generation model training from scratch with competitive performance. Moreover, we extend our YOSO to one-step text-to-image generation based on pre-trained models by several effective training techniques (i.e., latent perceptual loss and latent discriminator for efficient training along with the latent DMs; the informative prior initialization (IPI), and the quick adaption stage for fixing the flawed noise scheduler). Experimental results show that YOSO achieves the state-of-the-art one-step generation performance even with Low-Rank Adaptation (LoRA) fine-tuning. In particular, we show that the YOSO-PixArt-$\alpha$ can generate images in one step trained on 512 resolution, with the capability of adapting to 1024 resolution without extra explicit training, requiring only ~10 A800 days for fine-tuning. Our code is available at: https://github.com/Luo-Yihong/YOSO.

## 1 Introduction

Diffusion models (DMs) (Sohl-Dickstein et al., 2015; Ho et al., 2020; Song et al., 2021) have recently emerged as a powerful class of generative models, demonstrating state-of-the-art results in many generative modeling tasks, such as text-to-image (Rombach et al., 2022; Xu et al., 2023c; Chen et al., 2024; Feng et al., 2023), text-to-video (Blattmann et al., 2023; Hong et al., 2022), image editing (Hertz et al., 2022; Brooks et al., 2023; Meng et al., 2022) and controlled generation (Zhang et al., 2023; Mou et al., 2023). However, the generation process of DMs requires iterative denoising, leading to slow generation speed. Coupled with the intensive computational requirements of large-scale DMs, they constitute a substantial barrier to their practical application and wider adoption.

Sampling from DMs can be regarded as solving a probability flow ordinary differential equation (PF-ODE) (Song et al., 2021). Some previous works (Song et al., 2020; Lu et al., 2022a;b; Bao et al., 2022) focus on developing advanced ODE-solvers, for reducing the sampling steps. However, they still require 20+ steps to achieve high-quality generation. Another line is distilling from pre-trained PF-ODEs (Song et al., 2023; Liu et al., 2023; Luo et al., 2023a;b;c; Salimans & Ho, 2022), aiming to predict multi-step solution of PF-ODE solver by one step. Existing methods (Luo et al., 2023a;b) can achieve reasonable sample quality with 4+ steps. However, it is still challenging to generate high-quality samples with one step.

In contrast, generative adversarial networks (GANs) (Goodfellow et al., 2014; Radford et al., 2016) are naturally built on one-step generation with fast sampling speed. However, it is hard to extend GANs on large-scale datasets due to training challenges (Sauer et al., 2022; Kang et al., 2023), resulting in worse sample quality compared to DMs (Sauer et al., 2023b; Kang et al., 2023). In this

---

*Corresponding Author: Jing Tang (jingtang@ust.hk).

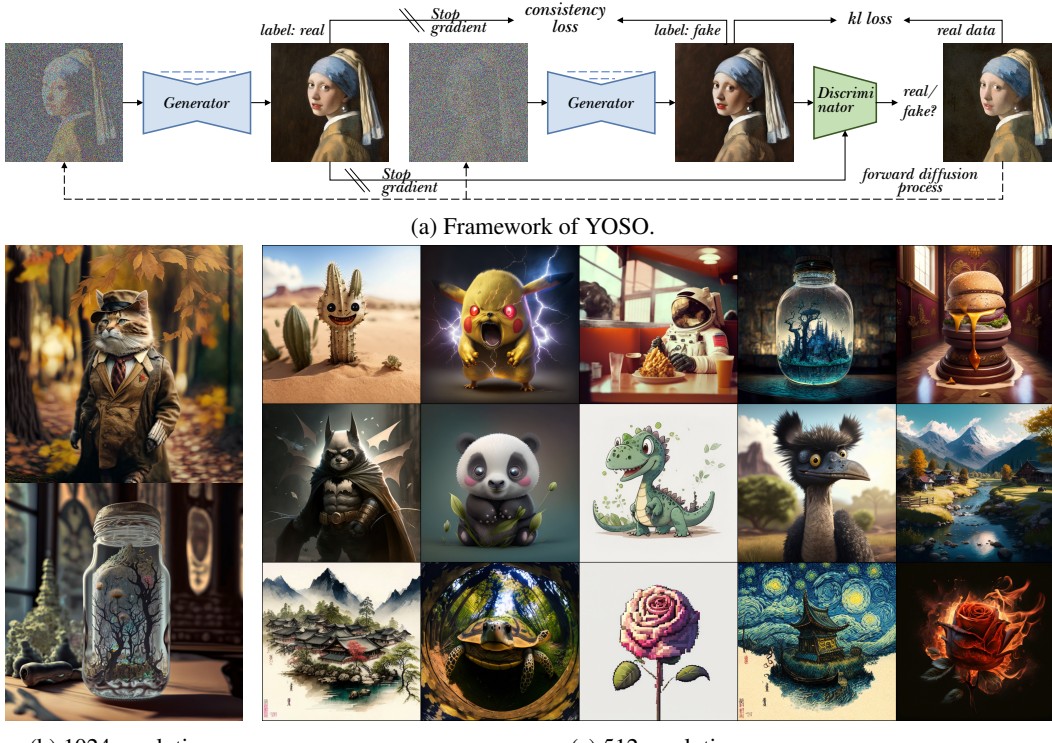

(a) Framework of YOSO.

(b) 1024 resolution.  (c) 512 resolution.

Figure 1: **One-step** generated images by YOSO under different configurations (Bottom). The model is trained by fine-tuning PixArt-$\alpha$ (Chen et al., 2024) on 512 resolution with our proposed algorithm. Bottom Left is generated by YOSO adapting to 1024 resolution with Eq. (7) without extra explicit training.

work, we present a novel approach to combine diffusion process and GANs. The key to achieving access is finding a way to smooth the adversarial divergence for stabilize the training while maintaining effective one-step learning. Previous works (Xiao et al., 2022; Xu et al., 2023b;c; Sauer et al., 2023c) have developed some variants for combining diffusion and GANs. However, existing works either directly perform adversarial divergence against real data without smoothing strategy which suffers from unstable training and mode collapse, or rely on adding noise to smooth the adversarial divergence to stabilize the training which delivers less effective one-step generation learning. To enjoy the best of both worlds, we propose to smooth the adversarial divergence by denoising the generator itself. In particular, we regard the one-step denoising generation based on less corrupted samples as ground truth and regard the one-step denoising generation based on more corrupted samples as student distribution to perform adversarial divergence. The approach can not only naturally reduce the distance between target distribution and student distribution to stabilize the training, but also form effective one-step learning on clean samples. The learning process can be viewed as a self-cooperative process (Xie et al., 2018), as the generator learns from itself. Such an innovative design enables stable training and effective learning for one-step generation. Hence, we name our model YOSO, short for You Only Sample Once.

Moreover, we extend our YOSO to one-step text-to-image generation based on pre-trained models and introduce several effective training techniques (i.e., latent perceptual loss and latent discriminator for efficient training along with the latent DMs; the informative prior initialization (IPI), and the quick adaption stage for fixing the noise scheduler). Thanks to our effective designs, we can efficiently and effectively fine-tune existing pre-trained text-to-image DMs (i.e., stable diffusion (Rombach et al., 2022) and PixArt-$\alpha$ (Chen et al., 2024)) for high-quality one-step generation (see Fig. 1). Furthermore, we are the pioneers in supporting Low-Rank Adaptation (LoRA) (Hu et al., 2022) fine-tuning in one-step text-to-image generation for enhanced efficiency, delivering state-of-the-art performance.

Our work presents several significant contributions:

- We introduce YOSO, a novel generative model that can generate high-quality images with one-step inference, with a stable training process and good mode coverage.
- We further scale up YOSO by several principled and effective proposed training techniques that enable low-resource fine-tuning of pre-trained text-to-image DMs for one-step text-to-image, requiring only ~10 A800 days.
- We conduct extensive experiments to demonstrate the effectiveness of the proposed YOSO, including image generation from scratch, text-to-image generation fine-tuning, compatibility with existing image customization and image controllable modules.

## 2 BACKGROUND

**Diffusion models.** Diffusion models (DMs) (Sohl-Dickstein et al., 2015; Ho et al., 2020) define a forward process that gradually transforms samples from data distribution to Gaussian distribution by adding noise in $T$ steps with variance schedule $\beta_t$: $q(\mathbf{x}_t|\mathbf{x}_{t-1}) \triangleq \mathcal{N}(\mathbf{x}_t; \sqrt{1-\beta_t}\mathbf{x}_{t-1}, \beta_t\mathbf{I})$. The corrupted samples can be directly obtained by $\mathbf{x}_t = \bar{\alpha}_t\mathbf{x}_0 + \sqrt{1-\bar{\alpha}_t}\epsilon$, where $\bar{\alpha}_t = \prod_{s=1}^t (1-\beta_T)$ and $\epsilon \sim \mathcal{N}(0, \mathbf{I})$. The parameterized reversed diffusion process is defined to gradually denoise: $p_\theta(\mathbf{x}_{t-1}|\mathbf{x}_t) \triangleq \mathcal{N}(\mathbf{x}_{t-1}; \mu_\theta(\mathbf{x}_t, t), \sigma_t^2\mathbf{I})$. The model can be trained by minimizing the negative ELBO (Ho et al., 2020; Kingma et al., 2021): $\mathcal{L} = \mathbb{E}_{t,q(\mathbf{x}_0)q(\mathbf{x}_t|\mathbf{x}_0)}\mathrm{KL}(q(\mathbf{x}_{t-1}|\mathbf{x}_t, \mathbf{x}_0)||p_\theta(\mathbf{x}_{t-1}|\mathbf{x}_t))$ where $q(\mathbf{x}_{t-1}|\mathbf{x}_t, \mathbf{x}_0)$ is Gaussian posterior derived in (Ho et al., 2020). A key assumption in diffusion is that the denoising step size from $t$ to $t-1$ is sufficiently small. This assumption ensures the true $q(\mathbf{x}_{t-1}|\mathbf{x}_t)$ follows Gaussian distribution, enabling the effectiveness of modeling $p_\theta(\mathbf{x}_{t-1}|\mathbf{x}_t)$ with Gaussian distribution.

**Diffusion-GAN hybrids.** An issue in DMs is that the true $q(\mathbf{x}_{t-1}|\mathbf{x}_t)$ does not follow Gaussian distribution when the denoising step size is not sufficiently small. Therefore, in order to enable large denoising step size, Diffusion GANs (Xiao et al., 2022) propose to minimize the adversarial divergence between model $p_\theta(\mathbf{x}'_{t-1}|\mathbf{x}_t)$ and $q(\mathbf{x}_{t-1}|\mathbf{x}_t)$: $\min_\theta \mathbb{E}_{q(\mathbf{x}_t)}[D_{\mathrm{adv}}(q(\mathbf{x}_{t-1}|\mathbf{x}_t)||p_\theta(\mathbf{x}'_{t-1}|\mathbf{x}_t))]$, where $p_\theta(\mathbf{x}'_{t-1}|\mathbf{x}_t) \triangleq \int p_\theta(\mathbf{x}_0|\mathbf{x}_t)q(\mathbf{x}_{t-1}|\mathbf{x}_t, \mathbf{x})d\mathbf{x}_0$ and $p_\theta(\mathbf{x}_0|\mathbf{x}_t)$ is imposed by a GAN generator. The capability of a GAN-based formulation enables much larger denoising step sizes (i.e., 4 steps).

## 3 METHOD: SELF-COOPERATIVE DIFFUSION GANs

A key issue in Diffusion-GAN hybrid models (Xiao et al., 2022; Xu et al., 2023b;c) is that they match the generator distribution $p_\theta(\mathbf{x}_{t-1}|\mathbf{x}_t) \triangleq \mathbb{E}_{p_\theta(\mathbf{x}_0|\mathbf{x}_t)}q(\mathbf{x}_{t-1}|\mathbf{x}_t, \mathbf{x}_0)$ with the corrupted data distribution. This formulation only indirectly learns the $p_\theta(\mathbf{x}_0|\mathbf{x}_t)$ and $p_\theta(\mathbf{x}_0) = \int q(\mathbf{x}_t)p_\theta(\mathbf{x}_0|\mathbf{x}_t)d\mathbf{x}_t$, which are the distributions used for one-step generation, making the learning process less effective.

### 3.1 OUR DESIGN

To enable more effective learning for one-step generation, we propose to directly construct the learning objectives over clean data. We first construct a sequence distribution of clean data as follows:

$$p_\theta^{(t)}(\mathbf{x}_0) = \int q(\mathbf{x}_t)p_\theta(\mathbf{x}_0|\mathbf{x}_t)d\mathbf{x}_t, \quad 0 < t \leq T; \quad p_\theta^{(0)}(\mathbf{x}_0) \triangleq q(\mathbf{x}_0), \tag{1}$$

where $q(\mathbf{x}_0)$ is the data distribution, $p_\theta(\mathbf{x}_0|\mathbf{x}_t) \triangleq \mathcal{N}(G_\theta(\mathbf{x}_t, t), \sigma^2\mathbf{I})$ and $G_\theta$ is the denoising generator. Note that $G_\theta(\mathbf{x}_t, t)$ is our denoising generator that predicts clean samples. If the network $\epsilon_\theta$ is parameterized to predict noise, we have $G_\theta(\mathbf{x}_t, t) \triangleq \frac{\mathbf{x}_t - \sqrt{1-\bar{\alpha}_t}\epsilon_\theta(\mathbf{x}_t, t))}{\bar{\alpha}_t}$.

Given the constructed distribution, we can formulate the optimization objective as follows:

$$\begin{aligned} &\mathbb{E}_t[D_{\mathrm{adv}}(q(\mathbf{x})||p_\theta^{(t)}(\mathbf{x})) + \lambda \cdot \mathrm{KL}(q(\mathbf{x}_0, \mathbf{x}_t)||p_\theta(\mathbf{x}_0, \mathbf{x}_t))] \\ &= \mathbb{E}_t[D_{\mathrm{adv}}(q(\mathbf{x})||p_\theta^{(t)}(\mathbf{x})) + \lambda_t \cdot \mathrm{KL}(q(\mathbf{x}_0)q(\mathbf{x}_t|\mathbf{x}_0)||q(\mathbf{x}_t)p_\theta(\mathbf{x}_0|\mathbf{x}_t))], \end{aligned} \tag{2}$$

where $q(\mathbf{x}_0, \mathbf{x}_t) \triangleq q(\mathbf{x}_0)q(\mathbf{x}_t|\mathbf{x}_0)$ and $p_\theta(\mathbf{x}_0, \mathbf{x}_t) \triangleq q(\mathbf{x}_t)p_\theta(\mathbf{x}_0|\mathbf{x}_t)$. The optimization objective is constructed by combining adversarial divergence and KL divergence. Specifically, the adversarial divergence focuses on matching over the distribution level, ensuring the generation quality, and the KL divergence focuses on matching over point level, ensuring the mode coverage.

However, it is hard to directly learn the adversarial divergence over clean data distribution, akin to the challenges with GAN training. To tackle these challenges, previous Diffusion GANs (Xiao et al., 2022; Xu et al., 2023b) have pivoted towards learning the adversarial divergence over the corrupted

data distribution. Unfortunately, as analyzed before, such an approach fails to directly match $p_\theta(\mathbf{x}_0)$, curtailing the efficacy of one-step generation. Moreover, it also compels the discriminator to fit different levels of noise, leading to limited capability.

Recall that $p_\theta^{(t)}(\mathbf{x})$ is defined as $\int q(\mathbf{x}_t)p_\theta(\mathbf{x}|\mathbf{x}_t)d\mathbf{x}_t$. The quality of the distribution has two key factors: 1) the ability of the trainable generator $G_\theta$; 2) the information given by $\mathbf{x}_t$.

Hence given the generator $G_\theta$ fixed, if we increase the information in $\mathbf{x}_t$, it is supposed that we can get a better distribution. In other words, it is highly likely that $p_\theta^{(t_k)}(\mathbf{x})$ is superior to $p_\theta^{(t)}(\mathbf{x})$, where $t_k = \max\{t - k, 0\} < t$. Motivated by the cooperative approach (Xie et al., 2018; 2021; 2022; Luo et al., 2024a) which uses an MCMC revised version of model distribution to learn the generator, we suggest to use $p_\theta^{(t_k)}(\mathbf{x})$ as ground truth to learn $p_\theta^{(t)}(\mathbf{x})$, which constructs the following training objective:

$$\min_\theta \mathcal{L}_\theta \triangleq \mathbb{E}_t \mathbb{E}_{q(\mathbf{x})q(\mathbf{x}_t|\mathbf{x})} \lambda_t ||G_\theta(\mathbf{x}_t, t) - \mathbf{x}||_2^2 + \mathbb{E}_t(D_{\mathrm{adv}}(p_\theta^{(t_k)}(\mathsf{sg}(\mathbf{x}))||p_\theta^{(t)}(\mathbf{x})), \tag{3}$$

where $\mathsf{sg}[\cdot]$ denotes stop-gradient operator and the second term is named as *cooperative adversarial loss*. This training objective can be regarded as a self-cooperative approach, since the 'revised' samples from $p_\theta^{(t_k)}(\mathbf{x})$ and samples from $p_\theta^{(t)}(\mathbf{x})$ are generated by the same network $G_\theta$. Note that we only replace data distribution as $p_\theta^{(t_k)}(\mathbf{x})$ in the adversarial divergence for smoothing the learning objective, as recent work (Luo et al., 2024b) has found that it is beneficial to learn the generator with a mix of real data and revised data.

We briefly verify the theoretical rationality of the cooperative adversarial divergence below.

**Proposition 1** *The optimal solution of the cooperative adversarial loss reaches $p_\theta^{(T)}(\mathbf{x}) = p_d(\mathbf{x})$.*

This proposition tells that the proposed cooperative adversarial loss can recover the real data distribution when the network's capability is sufficiently large, demonstrating the theoretical rationality of the proposed objective. See proof in the Appendix C.

In the distribution matching objective above, we apply the non-saturating GAN objective to minimize the adversarial divergence of marginal distributions. And the KL divergence for point matching can be optimized by $L_2$ loss. Hence a tractable training objective is formulated as follows:

$$\min_\theta \max_\phi \mathbb{E}_t[\mathbb{E}_{p_\theta^{(t_k)}(\mathbf{x})} \log D_\phi(\mathsf{sg}(\mathbf{x}), t) - \mathbb{E}_{p_\theta^{(t)}(\mathbf{x})} \log D_\phi(\mathbf{x}, t)] + \lambda_t \mathbb{E}_{q(\mathbf{x})q(\mathbf{x}_t|\mathbf{x})} ||G_\theta(\mathbf{x}_t, t) - \mathbf{x}||_2^2,$$

where $D_\phi$ is the discriminator network. We find that the self-cooperative approach is connected with Consistency Training (Song et al., 2023). However, Consistency Training considers the $\mathbf{x}_{t_k}$ as an approximated ODE solution of $\mathbf{x}_t$ to perform a point-to-point match. In contrast, our proposed objective matches $p_\theta^{(t)}(\mathbf{x})$ and $p_\theta^{(t_k)}(\mathbf{x})$ in marginal distribution level, which avoids the approximated error of ODE.

To further ensure the mode cover of the proposed model, we can add consistency loss to our objective as regularization, which constructs the following loss:

$$\min_\theta \max_\phi \mathbb{E}_t \big\{ \mathbb{E}_{p_\theta^{(t_k)}(\mathbf{x})}[\log D_\phi(\mathbf{x}, t) - \mathbb{E}_{p_\theta^{(t)}(\mathbf{x})} \log D_\phi(\mathbf{x}, t)]$$
$$+ \mathbb{E}_{q(\mathbf{x})q(\mathbf{x}_{t_k}|\mathbf{x})q(\mathbf{x}_t|\mathbf{x}_{t_k}, \mathbf{x})}[\lambda(t)||G_\theta(\mathbf{x}_t, t) - \mathbf{x}||_2^2 + \lambda_t^{\mathrm{con}}||G_\theta(\mathbf{x}_t, t) - \mathsf{sg}(G_\theta(\mathbf{x}_{t_k}, t_k))||_2^2] \big\}, \tag{4}$$

where $\lambda_t^{\mathrm{con}}$ and $\lambda(t)$ are pre-defined hyper-parameters.

# 4 TRY IT ON CIFAR-10 BEFORE SCALING UP FOR SAVING MONEY!

In this section, we evaluate the performance of the proposed YOSO on CIFAR-10 (Yu et al., 2015) to verify its effectiveness under both training from scratch and fine-tuning settings.

## 4.1 TRAINING STRATEGIES

Before starting training, we introduce some effective training strategies in following for taming YOSO.

**Decoupled scheduler.** We find that the optimal scheduler for performing consistency loss and adversarial loss are not identical. This is due to the cooperative adversarial loss not involving the approximated error regarding the timestep skips, enabling a substantial skip to maximize its efficacy.

In contrast, the consistency loss is susceptible to approximated error from timestep skips, necessitating a more conservative skip to preserve its effectiveness. Hence, to better unleash the capability of each loss, we propose using decoupled schedulers to construct our final training objective:

$$
\min_{\theta} \max_{\phi} \mathbb{E}_t \big\{ \mathbb{E}_{p_{\theta}^{(t_k)}(\mathbf{x})} [\log D_{\phi}(\mathbf{x}, t) - \mathbb{E}_{p_{\theta}^{(t)}(\mathbf{x})} \log D_{\phi}(\mathbf{x}, t)]
$$
$$
+ \mathbb{E}_{q(\mathbf{x})q(\mathbf{x}_{t_m}|\mathbf{x})q(\mathbf{x}_t|\mathbf{x}_{t_m},\mathbf{x})} [\lambda(t)||G_{\theta}(\mathbf{x}_t, t) - \mathbf{x}||_2^2 + \lambda_t^{\mathrm{con}}||G_{\theta}(\mathbf{x}_t, t) - \mathrm{sg}(G_{\theta}(\mathbf{x}_{t_m}, t_m))||_2^2] \big\},
\tag{5}
$$

where $t_k = \max(t - k, 0)$, $t_m = \max(t - m, 0)$, and we let $k = 250$ and $m = 25$ in experiments.

**Annealing strategy.** Since our target is to obtain a powerful one-step denoised generation model, however, as the KL loss and consistency loss will enforce point-level matching, this may hurt performance in cases where the model capacity is insufficient. Therefore, we suggest reducing the weight of these two losses to zero as training progresses. Specifically, we let $\lambda = (1 - \frac{\lfloor n/\frac{N}{K} \rfloor}{K-1})\lambda'$ for decreasing the weight to zero with $K$ times, where $n$ is the current iterations and $N$ is the total training iterations.

### 4.2 EMPIRICAL EVALUATION

**Experiment Setting.** We use the EDM (Karras et al., 2022) architecture for both UNet and discriminator. For the evaluation metric, we choose the Fréchet inception distance (FID) (Heusel et al., 2017). We train the proposed model on the CIFAR-10 dataset (Yu et al., 2015). We only employ MSE to compute image distance.

Table 1: Unconditional generation results on CIFAR-10.

| Model | FID↓ | NFE↓ |
|---|---|---|
| *Distillation or Fine-tuning* | | |
| **YOSO** | **1.81** | 1 |
| Tract (Berthelot et al., 2023) | 3.78 | 1 |
| Diff-Instruct (Luo et al., 2023c) | 4.53 | 1 |
| DMD (Yin et al., 2023) | 2.66 | 1 |
| CTM (Kim et al., 2024) | 1.98 | 1 |
| SiD (Zhou et al., 2024) | 1.92 | 1 |
| *Training from scratch* | | |
| **YOSO** | 2.26 | 1 |
| DDGANs (Xiao et al., 2022) | 3.75 | 4 |
| CT (Song et al., 2023) | 8.70 | 1 |
| iCT (Song & Dhariwal, 2024) | 2.83 | 1 |
| DDPM (Ho et al., 2020) | 3.21 | 1000 |
| EDM (Karras et al., 2022) | **2.04** | 36 |
| DDIM (Song et al., 2020) | 4.67 | 50 |
| StyleGAN2 (Karras et al., 2019) | 2.92 | 1 |

**Results.** The quantitative results are shown in Tab. 1. We observe that our method provides state-of-the-art performance with only a one-step manner on both training from scratch and fine-tuning settings, outperforming previous existing accelerated DMs, and GANs. In particular, our YOSO in fine-tuning EDM (Karras et al., 2022) even delivers better performance compared to EDM itself with ODE sampling. This is because our YOSO performs fine-tuning instead of distillation, the optimal solution for our YOSO is data distribution instead of teacher distribution.

## 5 TOWARDS ONE-STEP TEXT-TO-IMAGE SYNTHESIS

Since training a text-to-image model from scratch is quite expensive, we suggest using pre-trained text-to-image DMs as initialization with Self-Coopeartive Diffusion GANs. In this section, we introduce several principled designs for developing a generation model that enables one-step text-to-image synthesis based on pre-trained DMs.

### 5.1 USING PRE-TRAINED MODELS FOR TRAINING

**Latent Perceptual Loss.** Prior research (Hou et al., 2017; Hoshen et al., 2019; Song et al., 2023) has confirmed the effectiveness of perceptual loss in various domains. Notably, recent studies (Liu et al., 2023; Song et al., 2023) have found that the LPIPS loss (Zhang et al., 2018) is crucial for obtaining few-step DMs with high sample quality. However, a notable drawback is that the LPIPS loss is computed in the data space, which is expensive. In contrast, the popular SD operates in the latent space to reduce computational demands. Hence, using LPIPS loss in training latent DMs is pretty expensive, as it requires not only the computation of LPIPS loss in the data space but also an additional decoding operation. Realizing that the pre-trained SD can serve as an effective feature extractor (Xu et al., 2023a), we suggest using pre-trained SD to perform latent perceptual loss. However, SD is a UNet, whose final layer predicts epsilon with dimensions identical to the data. Hence we propose using the bottleneck layer of the UNet for computing:

$$
d(\mathbf{z}_{\theta}, \mathbf{z}) = ||\mathrm{HalfUNet}(\mathbf{z}_{\theta}, c, t = 0) - \mathrm{HalfUNet}(\mathbf{z}, c, t = 0)||_2^2,
\tag{6}
$$

where $\mathbf{z}$ is the latent encoded images by VAE and $c$ is the text feature. We note that the benefit of computing latent perceptual loss by SD is not only the computational efficiency but also the incorporation of text features, which are crucial in text-to-image tasks.

**Latent Discriminator.** Training GANs for the text-to-image on large-scale datasets faces serious challenges. Specifically, unlike unconditional generation, the discriminator for the text-to-image

task should justify based on both image quality and text alignment. This challenge is more obvious during the initial stage of training. To address this issue, previous pure GANs (Kang et al., 2023) for text-to-image propose complex learning objectives and require expensive costs for training. The learning of GANs has been shown to benefit from using a pre-trained network as the discriminator. As discussed above, the pre-trained SD has learned representative features. Hence, we suggest applying the pre-trained SD for constructing the latent discriminator. Similar to latent perceptual loss, we only use half UNet for the discriminator followed by a simple predict head. The advantages of the proposed strategy are twofold: 1) we use the informative pre-trained network as initialization; 2) the discriminator is defined over latent space, which is computationally efficient. Unlike previous work (Sauer et al., 2023c) that defines a discriminator over data space which required decode latent and backward from decoder, yielding expensive computational cost. By applying the Latent Discriminator, we observe a stable training process with fast convergence.

## 5.2 FIXING THE NOISE SCHEDULER

A common issue in DMs is that the final corrupted samples are not pure noise. For example, the noise scheduler used by the SD makes the corrupted samples at the final timestep as: $\mathbf{x}_T = 0.068265 \cdot \mathbf{x}_0 + 0.99767 \cdot \epsilon$, where the terminal Signal-to-noise ratio (SNR) is $\frac{\bar{\alpha}_T}{1-\bar{\alpha}_T} = 0.004682$, effectively creates a gap between training and inference. Previous work (Lin et al., 2024a) has only observed that this makes DMs unable to produce pure black or white images. However, we find that this issue yields serious problems in one-step generation. As shown in Fig. 2, there are notable artifacts in one-step generation if we directly sample noise from standard Gaussian. The reason may be that in multi-step generation, the gap can be gradually fixed in sampling, while one-step generation reflects the gap more directly. To address this issue, we provide two simple yet effective solutions.

**Informative Prior Initialization (IPI).** The non-zero terminal SNR issue is similar to prior hole issues in VAEs (Klushyn et al., 2019; Bauer & Mnih, 2019; Kingma et al., 2016). Therefore, we can use informative prior instead of non-informative prior to effectively address this issue. For simplicity, we adopt a learnable Gaussian Distribution $\mathcal{N}(\boldsymbol{\mu}, \sigma^2 \mathbf{I})$, whose optimal formulation is given below: $\epsilon'' = \bar{\alpha}_T \cdot (\mathbb{E}_{\mathbf{x}} \mathbf{x} + \text{Std}(\mathbf{x}) \times \epsilon') + \sqrt{1 - \bar{\alpha}_T} \cdot \epsilon$, where $\mathbb{E}_{\mathbf{x}} \mathbf{x}$ and $\text{Std}(\mathbf{x})$ can be efficiently estimated by finite samples, and $\epsilon'$ follows standard Gaussian distribution. As shown in Fig. 2, the artifacts in one-step generation are immediately removed after applying IPI. We note that the performance is achieved with minimal adjustment, enabling the possibility of developing one-step text-to-image generation by LoRA fine-tuning.

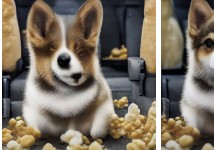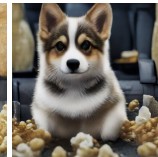

(a) w/o IPI     (b) w/ IPI

Figure 2: Samples by YOSO-LoRA with one-step inference from different initialization.

**Quick Adaption to V-prediction and Zero Terminal SNR.** The IPI suffers from numerical instability when terminal SNR is pretty low. As shown in Fig. 3, we fine-tune PixArt-$\alpha$ (Chen et al., 2024) whose terminal SNR is 4e-5 with introduced technique and $\epsilon$-prediction, failing in one-step generation. Following (Lin et al., 2024a), we suggest switching to v-prediction (Salimans & Ho, 2022) and zero terminal SNR. However, we found that direct transition convergences slowly (see Appendix F). This is unacceptable in solving large-scale text-to-image with limited computational resources. To address this, we propose a quick adapt-stage:

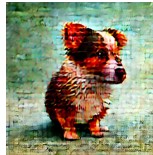

Figure 3: Predicting $\epsilon$ fails.

***Adapt-stage-I*** switch to v-prediction : $L(\theta) = \lambda_t ||v_\theta(\mathbf{x}_t, t) - v_\phi(\mathbf{x}_t, t)||_2^2$, where $v_\phi(\mathbf{x}_t, t) = \bar{\alpha}_t \epsilon_\phi(\mathbf{x}_t, t) - \sqrt{1 - \bar{\alpha}_t} \mathbf{x}_\phi^t$, $\mathbf{x}_\phi^t = \frac{\mathbf{x}_t - \sqrt{1-\bar{\alpha}_t}\epsilon_\phi(\mathbf{x}_t,t)}{\bar{\alpha}_t}$ , $v_\theta(\cdot, \cdot)$ denotes the desired v-prediction model, and $\epsilon_\phi(\cdot, \cdot)$ denotes the frozen pre-trained model.

***Adapt-stage-II*** switch to zero terminal SNR : $L(\theta) = \lambda_t ||v_\theta(\mathbf{x}_t, t) - v_\phi(\mathbf{x}_t', t)||_2^2$. Note that in this stage, we only change the scheduler to zero terminal SNR for student model. This avoids numerical instability issues of $\epsilon$-prediction with zero terminal SNR. We note that the zero terminal SNR scheduler has lower SNR than the original schedule at each timestep, formulating an effective distillation objective.

We note that this adapt stage converges rapidly, typically requiring only 1k iterations to initialize training YOSO. This enables a quick adaption to v-prediction and zero terminal SNR from pre-trained $\epsilon$-prediction DMs, thereby comprehensively mitigating the non-zero terminal SNR issue in the noise scheduler.

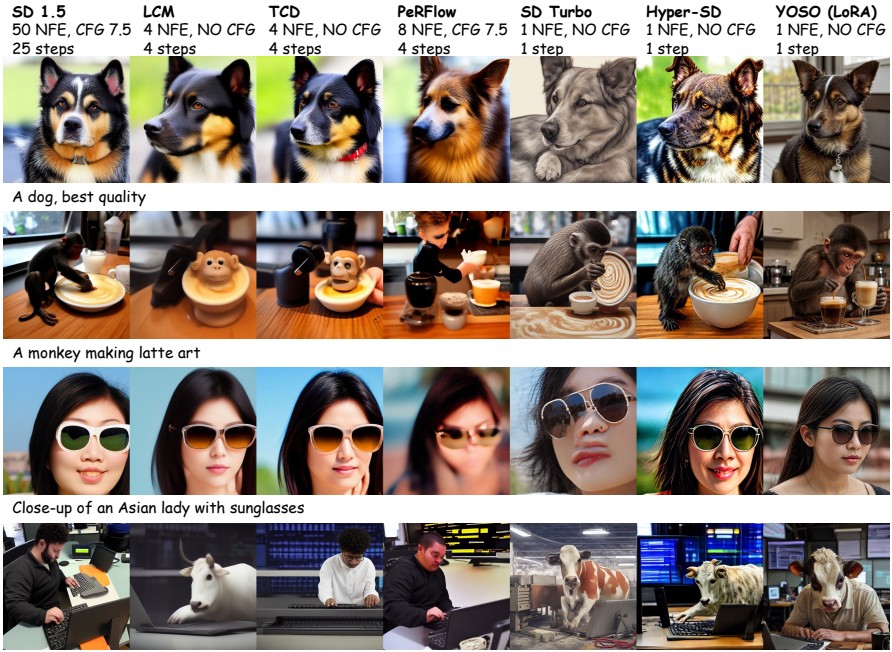

Figure 4: Qualitative comparisons of YOSO against competing methods. NFE denotes the Number of Function Evaluations.

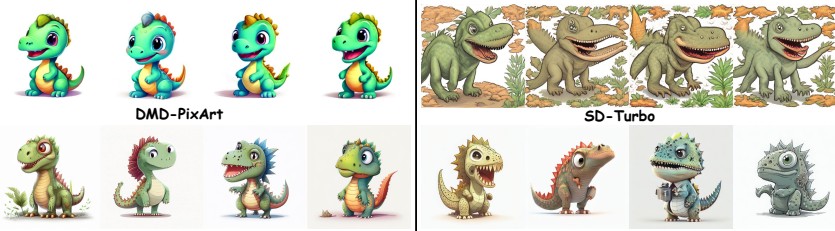

Figure 5: Qualitative comparisons of YOSO against competing methods. It can be seen that both DMD and SD-Turbo suffer from mode collapse, while YOSO achieves better sample quality and mode cover. The prompt is "A cute dinosaur, cartoon style, white background".

## 6 EXPERIMENTS

In this section, we evaluate the performance of the proposed YOSO. Sec. 6.1 examines YOSO in the context of text-to-image generation by fine-tuning pre-trained PixArt-$\alpha$ (Chen et al., 2024) and Stable Diffusion (Rombach et al., 2022). In Sec. 6.3, we show that the YOSO can be used for several downstream applications. Moreover, we conduct ablation studies in Sec. 6.2 to highlight the effectiveness of our proposed algorithm and proposed training techniques.

### 6.1 TEXT-TO-IMAGE GENERATION

**Experiment Setting.** We initialize the GAN generator by pre-trained PixArt-$\alpha$ which is a diffusion transformer with 0.6B parameters. For the GAN discriminator, we construct the latent discriminator by pre-trained SD 1.5 using the proposed construction introduced in Sec. 5.1. We switch the pre-trained PixArt-$\alpha$ to v-prediction by the proposed technique introduced in Sec. 5.1, followed by training on the JourneyDB dataset (Pan et al., 2023) with resizing as 512 resolution. We apply a batch-size of 256 and a constant learning rate of 2e-5 during training. We observe the training convergence fast, requiring only 30k iterations and around **10 A800 days** to be trained. We apply a batch-size of 256 and a constant learning rate of 2e-5 during training. We also conduct experiments on fine-tuning SD 1.5 via LoRA to show the effectiveness of the proposed YOSO.

**Evaluation.** We employ Aesthetic Score (AeS) (Schuhmann et al., 2022) to evaluate image quality and adopt the Human Preference Score (HPS) v2.1 (Wu et al., 2023) to evaluate the image-text alignment and human preference. The AeS is computed by an aesthetic score predictor trained on

LAION (Schuhmann et al., 2022) datasets, without considering the image-text alignment. HPS is trained to predict human preference given image-text pairs, considering the image-text alignment and human aesthetic. Additionally, we include ImageReward score (Xu et al., 2024), and CLIP score (Hessel et al., 2021) to provide a more comprehensive evaluation of the model performance. We mainly compare our model against the open-source state-of-the-art (SOTA) models, e.g., SD-Turbo (Sauer et al., 2023c), PixArt-DMD (Chen et al., 2024; Yin et al., 2023), and Hyper-SD (a concurrent work) (Ren et al., 2024).

**Quantitative Results.** The quantitative results are presented in Tab. 2. As shown in Tab. 2, YOSO clearly beat previous state-of-the-art methods across all metrics (including HPS, AeS, Image Reward score, and Clip Score). It is important to highlight that for the SD 1.5 backbone, we only use LoRA for fine-tuning SD 1.5, where only less than 10% parameters are tuned. Moroever, our method without human feedback learning (HFL) even outperforms Hyper-SD with HFL which potentially hacks the machine metric.

Table 2: Comparison of machine metrics on text-to-image generation across state-of-the-art methods. HFL denotes human feedback learning which might hack the machine metrics.

| Model | Backbone | HFL | Steps | HPS↑ | AeS↑ | IR↑ | CS↑ |
|---|---|---|---|---|---|---|---|
| Base Model | SD 2.1 | No | 25 | 27.28 | 5.66 | 0.36 | 33.46 |
| SD Turbo (Sauer et al., 2023c) | SD 2.1 | No | 1 | 27.06 | 5.31 | 0.40 | 32.21 |
| Base Model | SD 1.5 | No | 25 | 24.72 | 5.49 | 0.20 | 31.88 |
| InstaFlow (Liu et al., 2023) | SD 1.5 | No | 1 | 26.18 | 5.27 | -0.22 | 30.04 |
| Hyper-SD-LoRA (Ren et al., 2024) | SD 1.5 | Yes | 1 | 28.01 | 5.79 | 0.29 | 30.87 |
| YOSO-LoRA (Ours) | SD 1.5 | No | 1 | **28.33** | **5.97** | **0.43** | **31.33** |
| LCM-LoRA (Luo et al., 2023b) | SD 1.5 | No | 4 | 22.77 | 5.66 | -0.37 | 30.36 |
| PeRFlow (Yan et al., 2024) | SD 1.5 | No | 4 | 22.43 | 5.64 | -0.35 | 30.77 |
| TCD-LoRA (Zheng et al., 2024) | SD 1.5 | No | 4 | 22.24 | 5.45 | -0.15 | 30.62 |
| Hyper-SD-LoRA (Ren et al., 2024) | SD 1.5 | Yes | 4 | 30.24 | 5.55 | 0.53 | 31.07 |
| YOSO-LoRA (Ours) | SD 1.5 | No | 4 | **30.50** | **6.05** | **0.56** | **31.52** |
| Base Model | PixArt-α | No | 25 | 31.07 | 6.12 | 1.05 | 33.81 |
| DMD (Yin et al., 2023) | PixArt-α | No | 1 | 29.78 | 6.02 | 0.97 | 32.90 |
| YOSO (Ours) | PixArt-α | No | 1 | **30.52** | **6.19** | **1.02** | **32.95** |

**Qualitative Comparison.** To provide more comprehensive insight in understanding the performance of our YOSO, we further provide the qualitative comparison in Fig. 4. The comparison shows that: YOSO clearly beats existing baselines in terms of both image quality and prompt alignment, which is achieved by LoRA-finetuning. Specifically, the samples by SD-Turbo are blurry to some extent, and the samples by Hyper-SD are over-saturated and with notable artifacts. Moreover, the advantage of YOSO in text-image alignment is significant. For example, in the last row of Fig. 4 related to a cow-people worker is coding, previous methods including SD 1.5 itself cannot produce samples following the prompt, while YOSO precisely follows the prompt to generate a half cow half people worker. Additionally, it is important to highlight that both SD-Turbo and DMD seem to have a serious issue in *mode collapse*. As shown in Fig. 5, both SD-Turbo and DMD generate samples that are extremely close to each other. Overall, based on quantitative and qualitative results, we conclude that our YOSO is better in sample quality, prompt alignment, and mode cover compared with the SOTA one-step text-to-image models.

### 6.1.1 ZERO-SHOT ONE-STEP 1024 RESOLUTION GENERATION

Due to the computational resource limitation, we trained YOSO on 512 resolution. However, there is a 1024-resolution version of PixArt-$\alpha$, obtained by continuing training on the 512-resolution version of PixArt-$\alpha$. Motivated by the effectiveness of the LoRA combination (Luo et al., 2023b), we suggest constructing a similar combination as follows:

$$\mathbf{W}_{\text{YOSO}_{1024}} = \mathbf{W}_{\text{PixArt}_{512}} + \alpha \cdot (\mathbf{W}_{\text{PixArt}_{1024}} - \mathbf{W}_{\text{PixArt}_{512}}) + \beta \cdot (\mathbf{W}_{\text{YOSO}_{512}} - \mathbf{W}_{\text{PixArt}_{512}}), \quad (7)$$

where $\alpha, \beta \in (0, 1]$ and $\mathbf{W}$ means the model weight. We let $\alpha = \beta = 1$ in our experiments. As shown in Fig. 1b, the YOSO constructed in this way can generate high-quality images with 1024 resolution. Note that YOSO even changes the predicted objective to v-prediction. The impressive performance indicates the robust generalization ability of our proposed YOSO.

### 6.2 ABLATION STUDIES

We provide additional insight into the effectiveness of the proposed YOSO by performing ablation studies on some potential variants on CIFAR-10 and on text-to-image generation. We call YOSO without decoupled scheduler and annealing strategy as YOSO-base. See more details in Appendix.

**Effect of Consistency Loss.** For the effect of consistency loss in YOSO, our results suggest that it can improve the image quality, as indicated by the FID scores in Tab. 3. Notably, the removal of the consistency loss does not lead to a substantial decline in generation performance, which indicates the effectiveness of our proposed adversarial formulation.

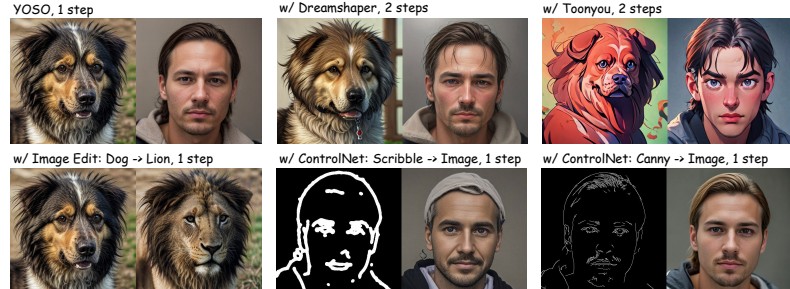

Figure 6: Qualitative comparison against competing methods and applications in down-stream tasks.

Table 3: Ablation study on CIFAR-10 with smaller backbone.

| Model | FID↓ | NFE↓ |
|---|---|---|
| **YOSO** | **3.05** | 1 |
| **YOSO-Base w/ decoupled scheduler** | 3.30 | 1 |
| **YOSO-Base** | 3.82 | 1 |
| YOSO-Base w/o consistency loss | 4.41 | 1 |
| YOSO-Base w/o LPIPS loss | 4.63 | 1 |
| YOSO-Base w/ $D_{\mathrm{adv}}(p_d||p_\theta^t)$ | 10.85 | 1 |
| UFOGen (Xu et al., 2023c) | 45.15 | 1 |
| DDGANs (Xiao et al., 2022) | 3.75 | 4 |

Table 4: Ablation study on one-step text-to-image synthesis.

| CM | Lper | IPI | Naive GAN | Coop GAN | DS | AS | HPS | AeS |
|---|---|---|---|---|---|---|---|---|
| ✓ | | | | | | | 15.22 | 5.23 |
| ✓ | ✓ | | | | | | 20.08 | 5.45 |
| ✓ | ✓ | ✓ | | | | | 20.63 | 5.57 |
| ✓ | ✓ | ✓ | ✓ | | | | 24.88 | 5.69 |
| ✓ | ✓ | ✓ | | ✓ | | | 26.42 | 5.85 |
| ✓ | ✓ | ✓ | | ✓ | ✓ | | 27.10 | 5.94 |
| ✓ | ✓ | ✓ | | ✓ | ✓ | ✓ | **28.33** | **5.97** |

**Effect of Latent Perceptual Loss.** To investigate the effect of latent perceptual loss proposed in Sec. 5.1, we conduct experiments on text-to-image tasks by using it to compute the consistency loss. As shown in Tab. 4, the proposed latent perceptual loss clearly improves the HPS from 15.22 to 20.08 and AeS from 5.23 to 5.45, highlighting its effectiveness.

**Effect of Decoupled Scheduler.** We evaluate the impact of the proposed decoupled scheduler by adding it to YOSO-base. As evidenced by Tab. 3 and Tab. 4, the incorporation of the decoupled scheduler significantly enhances sample quality, as reflected by improvements in FID, HPS, and AeS metrics. Notably, the FID score on CIFAR-10 decreases from 3.82 to 3.30 and the HPS in text-to-image tasks is improved from 26.42 to 27.10, demonstrating the superior efficacy of the decoupled scheduler.

**Effect of Annealing Strategy.** We also assess the effect of the proposed annealing strategy by removing it from the final formulated YOSO. As illustrated in Tab. 3 and Tab. 4, the removal of the annealing strategy degrades FID from 3.05 to 3.30 and reduces HPS from 28.33 to 27.10. Such results demonstrate the effectiveness of the annealing strategy in training YOSO.

**Adversarial Divergence** $D_{\mathrm{adv}}(p_d||p_\theta^t)$**.** The key design of our proposed YOSO is smoothing the adversarial divergence by using self-generated data instead of real data as ground truth to perform adversarial divergence. We evaluate the variant of using $D_{\mathrm{adv}}(p_d||p_\theta^t)$ as adversarial divergence. As shown in Tab. 3, the performance of the variant is significantly worse than YOSO-Base, i.e., FID degrades from 3.82 to 10.85. Moreover, besides sample quality, we find that the training of our proposed YOSO

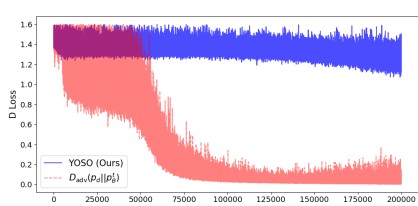

Figure 7: The discriminator loss curve.

is much more stable than the variant. As shown in Fig. 7, the discriminator loss of the variant is not stable and is much smaller than ours. We also compare the variant in the text-to-image task. The results are given in Tab. 4, from which we can see that replacing our cooperative adversarial loss with $D_{\mathrm{adv}}(p_d||p_\theta^t)$ yields serious performance deterioration, i.e., HPS decreases from 26.42 to 24.88 and AeS decreases from 5.85 to 5.69. This indicates that our approach can effectively smooth the adversarial divergence by narrowing the gap between fake and real distribution, which makes discrimination harder and hence increases discriminator loss. Furthermore, we observe mode collapse in the training of this variant. Note that a similar case also occurs in SD-Turbo, a strong one-step text-to-image model, which performs adversarial divergence directly against real data (See Fig. 4). Overall, these results indicate the superiority of the proposed YOSO.

**Comparison to UFOGen.** Similar to us, UFOGen does not directly perform adversarial divergence against real data either. Instead, they employ adversarial divergence over corrupted data (i.e., $D_{\mathrm{adv}}(q(\mathbf{x}_{t-1})|| \int q(\mathbf{x}_{t-1}|\mathbf{x} = G_\theta(\mathbf{x}_t, t))q(\mathbf{x}_t)d\mathbf{x}_t))$. However, we argue that such a formulation

is less effective in constructing one-step generation model as the adversarial divergence it employs only indirectly aligns with one-step generation at the level of clean data. We implement UFOGen on CIFAR-10 on our own. Note that we also perform consistency loss in the UFOGen variant to ensure a fair comparison, as the consistency loss has been found useful in YOSO. As shown in Tab. 3, UFOGen only obtains 45.15 FID on CIFAR-10, significantly worse than YOSO with 3.82 FID. This further underscores the superiority of our proposed YOSO. Through comparison with different adversarial divergence formulations, we find that our approach not only stabilizes training by smoothing the adversarial divergence but also facilitates effective learning for one-step generation.

### 6.3 Application

One promising and attractive property of DMs is that they can be used in multiple downstream tasks. In this section, we show the capability of our proposed YOSO in various downstream tasks, keeping the unique advantages of DMs: **1) Image-to-image Editing:** As shown in Fig. 6, our YOSO-LoRA is capable of performing high-quality image-to-image editing (Meng et al., 2022) in one step; **2) Compatibility with ControlNet and Different Base Models:** As shown in Fig. 6, our YOSO-LoRA is compatible with ControlNet (Zhang et al., 2023), following the condition well. Our YOSO-LoRA is also compatible with different base models (e.g., Dreamshaper and Toonyou) fine-tuned from SD 1.5, preserving their style well. We find that when applying YOSO-LoRA to new base models, it fails to produce reasonable samples in one step. This may be due to the capability of LoRA and the distribution shift of training data.

## 7 Related Work

**Few-Step Text-to-Image Generation.** LCM (Luo et al., 2023a) adapts consistency distillation (Song et al., 2023) for stable diffusion (SD) and enables image generation in 4 steps with acceptable quality. InstaFlow (Liu et al., 2023) adapts rectified flows (Liu et al., 2022; Liu, 2022) for SD, facilitating the generation of text-to-image in a mere single step. Despite this, the fidelity of the generated images is still poor. Some works (Yin et al., 2023) adapts variation score distillation (Wang et al., 2023b) and diff-instruct (Luo et al., 2023c) for SD, which requires training an additional DMs for the generator distribution, akin to the discriminator in GANs. We found models trained in this way seem to have serious mode collapse issues (see Fig. 5). Recently, combining DMs and GANs for one-step text-to-image generation has been explored. UFOGen (Xu et al., 2023c) extends DDGANs (Xiao et al., 2022) and SSIDMs (Xu et al., 2023b) to SD by modifying the computation of reconstruction loss from corrupted samples to clean samples. However, it still performs adversarial matching using corrupted samples. ADD (Sauer et al., 2023c) introduces a one-step text-to-image generation based on SD. It follows earlier research (Sauer et al., 2023a) by employing a pre-trained image encoder, DINOv2 (Oquab et al., 2024), as the backbone of the discriminator to accelerate the training. However, the discriminator design moves the training from latent space to pixel space, which substantially heightens computational demands. Moreover, they directly perform adversarial matches at clean real data, increasing the challenge of training. This requires the need for a better but expensive discriminator design and expensive R1 regularization to stabilize the training. SDXL-Lightning (Lin et al., 2024b) combines progressive distillation and adversarial training in a progressive framework. Recently, a concurrent work Hyper-SD (Ren et al., 2024) combines consistency distillation and adversarial training in a progressive framework, while our work can be trained end-to-end. However, their work performs adversarial point-level matching via consistency loss, and they still perform adversarial training via injecting noise. This results in less effective one-step learning, thus they rely on human feedback learning for achieving better performance. In contrast, we perform the adversarial training at the distribution level by replacing real data with self-generated data to smooth the adversarial divergence, which not only can stabilize the training without injecting noise but also form effective one-step generation learning. Moreover, compared with above mentioned diffusion-GAN hybrid models, our approaches can be trained from scratch to perform one-step generation, which is not demonstrated by them. Additionally, we extend our method not only to SD but also to PixArt-$\alpha$ (Chen et al., 2024) which is based on diffusion transformer (Peebles & Xie, 2022). This demonstrates the wide application of our proposed YOSO.

## 8 Conclusion

In summary, we present YOSO, a new generative model enabling high-quality one-step generation. Our novel designs combine the diffusion process and GANs, enabling not only one-step generation training from scratch but also one-step text-to-image generation fine-tuning from pre-trained models. We will release our model to advance the research of text-to-image synthesis.

ACKNOWLEDGMENTS

Jing Tang's work is partially supported by National Key R&D Program of China under Grant No. 2023YFF0725100 and No. 2024YFA1012701, by the National Natural Science Foundation of China (NSFC) under Grant No. 62402410 and No. U22B2060, by Guangdong Provincial Project (No. 2023QN10X025), by Guangdong Basic and Applied Basic Research Foundation under Grant No. 2023A1515110131, by Guangzhou Municipal Science and Technology Bureau under Grant No. 2023A03J0667 and No. 2024A04J4454, by Guangzhou Municipal Education Bureau (No. 2024312263), and by Guangzhou Municipality Big Data Intelligence Key Lab (No. 2023A03J0012), Guangzhou Industrial Information and Intelligent Key Laboratory Project (No. 2024A03J0628) and Guangzhou Municipal Key Laboratory of Financial Technology Cutting-Edge Research (No. 2024A03J0630).

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

---

**Algorithm 1** YOSO training from scratch.

---

**Require:** dataset $\mathcal{D}$, learning rate $\eta$, total denoising steps $T$, total iterations $N$, distance metric $d(\cdot, \cdot)$, and noise scheduler $\mathcal{Q}(\cdot, \cdot, \cdot)$.
**Ensure:** optimized models $G_\theta$, $D_\phi$.
 1: Initialize weights $\{\theta, \phi\}$;
 2: **for** $i \leftarrow 1$ **to** $N$ **do**
 3:     Sample noise $\epsilon$ from standard normal distribution;
 4:     Sample data $\mathbf{x}$ from dataset $\mathcal{D}$;
 5:     Sample Timesteps $t$ from 1 to $T$ uniformly.
 6:     Obtain noisy samples $\mathbf{x}_t = \mathcal{Q}(\mathbf{x}, \epsilon, t)$
 7:     Obtain less noisy samples $\mathbf{x}_{t-1} = \mathcal{Q}(\mathbf{x}, \epsilon, t-1)$
 8:     Obtain one-step prediction: $\hat{\mathbf{x}}^t = G_\theta(\mathbf{x}_t, t)$ and $\hat{\mathbf{x}}^{t-1} = G_\theta(\mathbf{x}_{t-1}, t-1)$
 9:     # update discriminator
10:     Compute Loss $\mathcal{L}_\phi$ following Eq. (4).
11:     $\phi \leftarrow \phi - \eta \nabla_\phi \mathcal{L}_\phi$;
12:     # update Generator
13:     Compute Loss $\mathcal{L}_\theta$ following Eq. (4).
14:     $\theta \leftarrow \theta - \eta \nabla_\theta \mathcal{L}_\theta$;
15: **end for**

---

## A   BROADER IMPACTS

This work presents YOSO, a method that accelerates multi-step large-scale text-to-image diffusion models into a one-step generator. On the positive side, while this is academic research, we believe the proposed YOSO can be widely applied in industry, and its high efficiency can lead to energy savings and environmental benefits. However, when these rapid generation models are manipulated by malicious actors, they can also simplify and accelerate the creation of harmful information. Although our work focuses on scientific research, we will take actions to reduce the harmful information, such as filtering out harmful content in the dataset.

## B   LIMITATION

Our model, like most text-to-image diffusion models, may exhibit shortcomings in terms of fairness, as well as in handling specific details and accurately controlling the number of targets. We plan to explore these unresolved issues in the generation field in our future work, in order to enhance the model's capabilities in text generation, fairness, detail control, and quantitative control.

## C   PROOF

*Proof:* The cooperative adversaril loss is defined as: $\mathbb{E}_t(D_{\mathrm{adv}}(p_\theta^{(t_k)}(\mathrm{sg}(\mathbf{x}))||p_\theta^{(t)}(\mathbf{x}))$. Since the divergence is non-negative, and the $p_\theta^{(0)}$ is defined as $p_d$, there exist a global optimal solution $p_\theta^{(T)}(\mathbf{x}) = p_\theta^{(T-1)}(\mathbf{x}) = \cdots = p_d(\mathbf{x})$ such that $\mathbb{E}_t(D_{\mathrm{adv}}(p_\theta^{(t_k)}(\mathrm{sg}(\mathbf{x}))||p_\theta^{(t)}(\mathbf{x})) = 0$. This completes the proof.

## D   TRAINING ALGORITHM

We present the algorithm for training YOSO from scratch in Algorithm 1.

## E   ADDITIONAL RELATED WORK

**Text-to-image Diffusion Models.** Since Diffusion models have shown a stable training process and are well-suited for scaling up generative models, numerous works have been proposed to extend DMs for text-to-image generation (Ramesh et al., 2022; Balaji et al., 2022; Rombach et al., 2022;

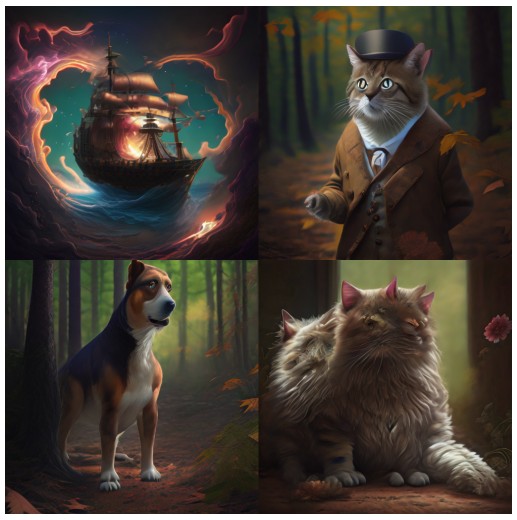 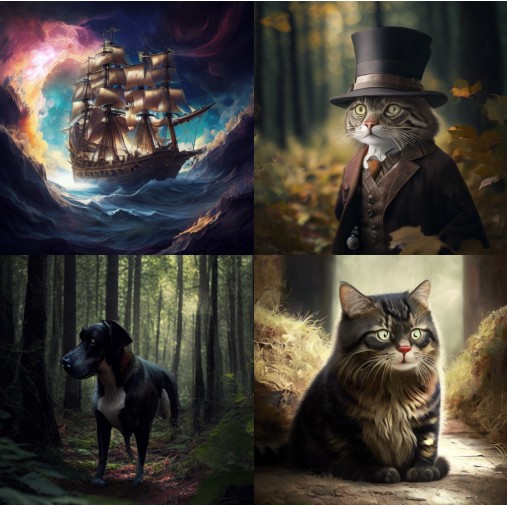

(a) w/o quick adaption.        (b) w/ quick adaption.

Figure 8: **One-step** generated images by YOSO-PixArt-$\alpha$ trained with only `5k iterations` under different configurations from the same initial noise and prompt.

Xue et al., 2023). Latent DMs (Rombach et al., 2022; Podell et al., 2023) are widely adopted for high-resolution image and video generation due to their computational efficiency.

## F  THE EFFECT OF QUICK-ADAPTION

We present a qualitative comparison between YOSO w/o quick adaption and YOSO w/ quick adaption in Fig. 8. Note that the YOSO and variants here are trained with only `5k` iterations. And for YOSO w/ quick adaption, we incorporate the `1k` iterations used for quick adaptation into the total `5k` training iterations to ensure a fair comparison. As shown in Fig. 8, YOSO w/ quick adaption demonstrates significantly better image quality compared to YOSO w/o quick adaption. The generated images of YOSO w/o quick adaption exhibit over-smoothing and severe artifacts. This indicates that the convergence of direct transition to v-prediction and zero terminal SNR is relatively slower. Additionally, it is worth highlighting that although there are only `5k` training iterations, the one-step generation quality of YOSO is already reasonable, which demonstrates the effectiveness and quick convergence of the proposed quick adaption stage and YOSO.

## G  EXPERIMENT SETTING DETAILS

### G.1  UNCONDITIONAL GENERATION EXPERIMENTS

For the generator, we use the Adam optimizer with $\beta_1 = 0.9$ and $\beta_2 = 0.999$; for the discriminator, we use the Adam optimizer with $\beta_1 = 0.$ and $\beta_2 = 0.999$. We adopt a constant learning rate of 2e-4 for both the discriminator and the generator in training from scratch. We adopt a constant learning rate of 2e-5 for both the discriminator and the generator in fine-tuning. We apply EMA with a coefficient of 0.9999 for the generator. We let the $\lambda_t = \text{SNR}(t)$ and $\lambda_t^{\text{con}} = \frac{1}{\frac{1}{\text{SNR}(t)} - \frac{1}{\text{SNR}(t-1)}}$.

### G.2  TEXT-TO-IMAGE EXPERIMENTS

For the generator, we use the AdamW optimizer with $\beta_1 = 0.9$ and $\beta_2 = 0.999$; for the discriminator, we use the AdamW optimizer with $\beta_1 = 0.$ and $\beta_2 = 0.999$. We adopt a constant learning rate of 2e-5 for both the discriminator and the generator. We apply gradient norm clipping with a value of 1.0 for the generator only. We use batch size 256. For full fine-tuning, we apply EMA with a coefficient of 0.9999 for the generator. For LoRA fine-tuning, we apply EMA with a coefficient of 0.999 for

Figure 9: Additional qualitative comparison on PixArt-$\alpha$ backbone at 1024 resolution.

the generator. Generally, the training is done within 30k iterations for full fine-tuning, while within 5k iterations for LoRA fine-tuning. We let the $\lambda_t = \text{SNR}(t)$ and $\lambda_t^{\text{con}} = \frac{1}{\frac{1}{\text{SNR}(t)} - \frac{1}{\text{SNR}(t-m)}}$.

For full fine-tuning, we train YOSO on the JourneyDB dataset (Pan et al., 2023), by resizing to 512 resolution. And we only use the square image. For LoRA fine-tuning, we use an internally collected dataset and the caption of the JourneyDB dataset (Pan et al., 2023) to generate data for training, we only generate one image for one caption. For the evaluation, we evaluate the HPS score on its benchmark, and we evaluate other metrics based on COCO-5k (Lin et al., 2014) datasets.

### G.3 Ablation Study

For the ablation study on CIFAR-10, we use the same UNet and discriminator architecture as used in DDGANs (Xiao et al., 2022). For the generator, we use the Adam optimizer with $\beta_1 = 0.9$ and $\beta_2 = 0.999$; for the discriminator, we use the Adam optimizer with $\beta_1 = 0.$ and $\beta_2 = 0.999$. We adopt a constant learning rate of 2e-4 for both the discriminator and the generator. We apply EMA with a coefficient of 0.9999 for the generator. We let the $\lambda_t = \text{SNR}(t)$ and $\lambda_t^{\text{con}} = \frac{1}{\frac{1}{\text{SNR}(t)} - \frac{1}{\text{SNR}(t-1)}}$.

## H  Additional Experiments

**Comparison to other Diffusion-Gan Hybrid Methods on High-Resolution Generation.** We conducted comparative experiments between methods by fine-tuning pre-trained diffusion models. We trained a lightweight diffusion model in the latent space of DC-AE (Chen et al., 2025) for efficiency. Specifically, following DC-AE, we adopted DiT-S as the model backbone, which is a lightweight model with 33M parameters. For a fair comparison, we initialized the generator for YOSO as well as the other methods using the pre-trained diffusion model. As shown in Tab. 6, YOSO achieves better performance on FFHQ-1024 compared to other diffusion-gan hybrid distillation methods (Xu et al., 2023c; Wang et al., 2023a; Kang et al., 2024).

Table 5: Training from scratch on ImageNet-64.

| Method | NFE | FID ($\downarrow$) |
|---|---|---|
| BigGAN-deep | 1 | 4.06 |
| StyleGAN-XL | 1 | 1.52 |
| ADM | 250 | 2.07 |
| EDM | 511 | 1.36 |
| CT | 1 | 13.0 |
| iCT | 1 | 3.25 |
| **YOSO (Ours)** | 1 | **2.65** |

Table 6: Comparison of different diffusion-gan hybrid methods on FFHQ-1024.

| Method | NFE | FID ($\downarrow$) |
|---|---|---|
| Teacher Diffusion (Our reproduced) | 250 | 13.88 |
| Diffusion-GAN | 1 | 22.16 |
| Diffusion2GAN | 1 | 18.01 |
| UFOGen | 1 | 24.37 |
| **YOSO (Ours)** | 1 | 16.23 |

**Training from scratch on ImageNet-64.** To further demonstrate the ability of YOSO in training from scratch, we conducted additional experiments on ImageNet-64 using the EDM architecture, demonstrating YOSO's effectiveness when trained from scratch. As shown in Tab. 5, our method achieves a better FID score compared to other methods, indicating both efficiency and improved performance.

**Zero-Shot COCO FID.** We argue that zero-shot COCO FID is not a good metric for evaluating text-to-image models. Existing work has found that zero-shot COCO FID can even be negatively correlated with human preferences in certain cases (Kirstain et al., 2023; Podell et al., 2023). We report the metric for completeness. We retrain the YOSO by fine-tuning on COYO-700M and our internal dataset. As shown in Tab. 7, YOSO is capable of achieving a low zero-shot COCO FID-30k of 8.90, outperforming prior text-to-image GANs.

| Model | Latency $\downarrow$ | FID-30k $\downarrow$ | CLIP Score-30k $\uparrow$ |
|---|---|---|---|
| StyleGAN-T | 0.10s | 13.90 | – |
| GigaGAN | 0.13s | 9.09 | – |
| SD v1.5 (CFG=7.5) | 2.59s | 13.45 | 0.32 |
| SD v1.5 (CFG=3) | 2.59s | 8.78 | 0.32 |
| LCM (4-step) | 0.26s | 23.62 | 0.30 |
| Diffusion2GAN | 0.09s | 9.29 | – |
| UFOGen | 0.09s | 12.78 | – |
| InstaFlow-0.9B | 0.09s | 13.27 | 0.28 |
| DMD | 0.09s | 14.93 | 0.32 |
| SwiftBrush | 0.09s | 16.67 | 0.29 |
| **YOSO** (full-finetuning) | 0.09s | **8.90** | 0.31 |

Table 7: Comparison of zero-shot COCO FID

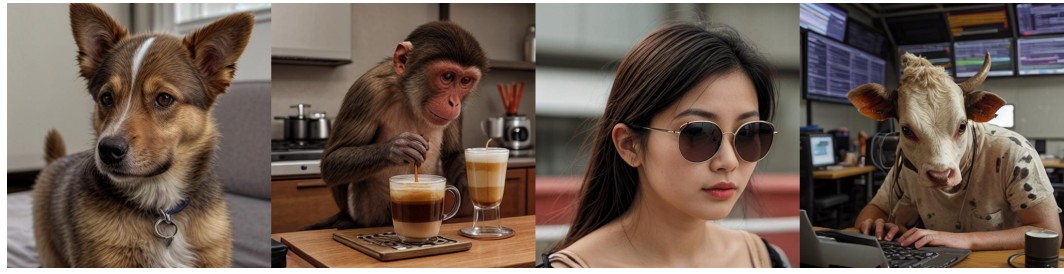

(a) 1-step samples

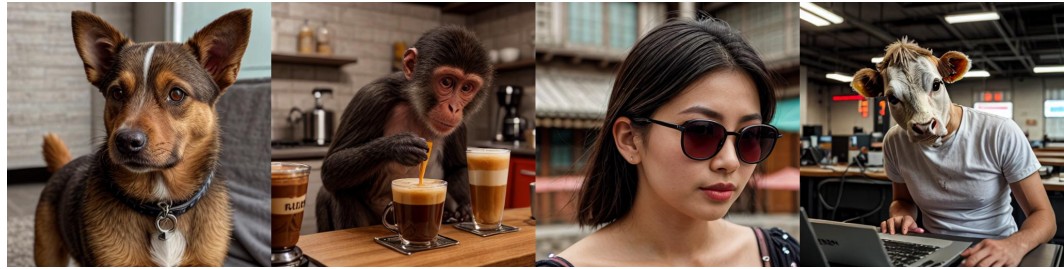

(b) 4-step samples

Figure 10: Qualitative comparison among different sampling steps using YOSO-LoRA.

# I ADDITIONAL QUALITATIVE COMPARISON

We present an Additional Qualitative comparison to PixArt-$\alpha$ and PixArt-$\delta$ on 1024 resolution in Fig. 9. As can be observed, our method produces significantly better image quality compared to PixArt-$\delta$ (LCM) and achieves comparable results to the multi-step teacher model.

We present an Additional qualitative comparison among different sampling steps using YOSO-LoRA in Fig. 10. As can be observed, the 4-step samples have better visual quality than 1-step samples

We present an Additional qualitative comparison among 2-step examples of LCM by varying the distance metrics in Fig. 11 to verify the effectiveness of the proposed latent perceptual loss. In particular, we mainly vary the feature layer for computing the latent perceptual loss. It can be seen that using the bottleneck layer delivers better visual quality. More importantly, **regardless of which layers are used to compute LPL, it is significantly better than MSE**. This highlights the effectiveness of our proposed latent perceptual loss.

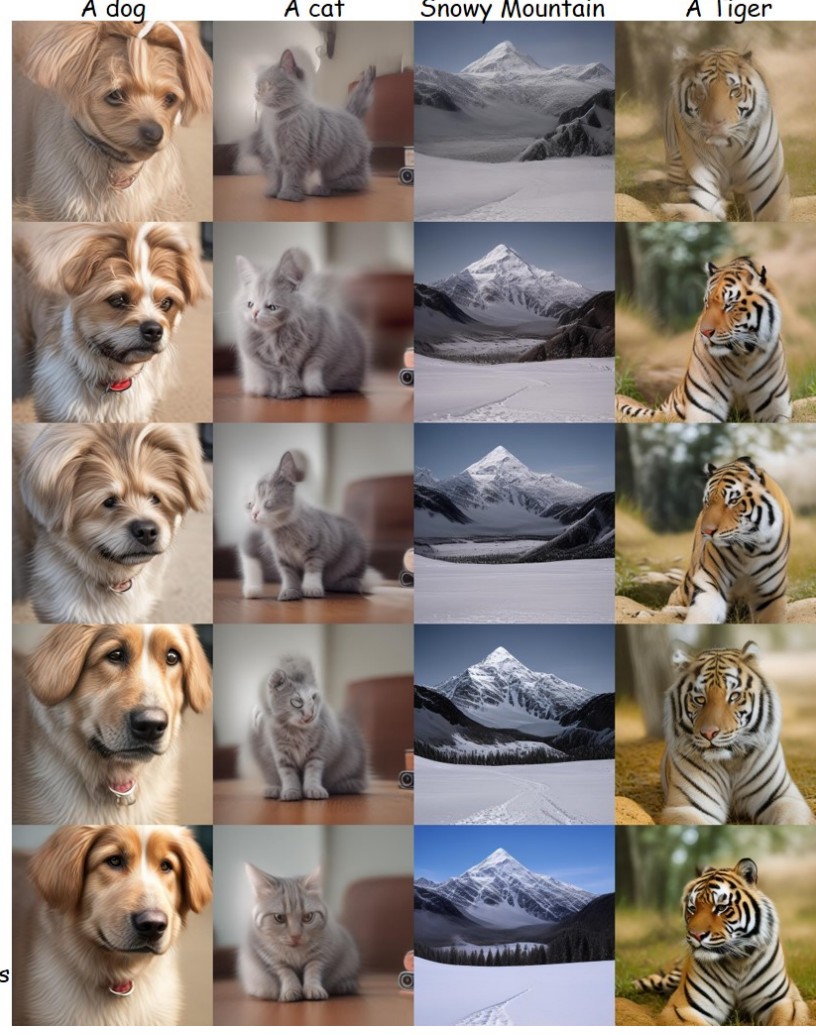

Figure 11: Additional qualitative comparison of 2-step examples on varying the distance metric in LCM.

