# OpenReview forum: "You Only Sample Once: Taming One-Step Text-to-Image Synthesis by Self-Cooperative Diffusion GANs"
_ICLR.cc/2025/Conference — ICLR 2025 Poster_

### Official Review · Reviewer_Awjz · 2024-10-27

**Soundness:** 2
**Presentation:** 3
**Contribution:** 2
**Rating:** 5
**Confidence:** 4

**Summary:**

The paper proposes a novel one-step generative model named YOSO, which offers higher training stability and model coverage compared to existing methods. This approach aims to achieve high-quality generation while maintaining performance metrics in a single step. Experiments are conducted on CIFAR-10 using HPS and AeS evaluation metrics.

**Strengths:**

1. Applying the Latent Discriminator leads to a stable training process with fast convergence.
2. The paper tackles a relevant application (i.e., ControlNet) with one-step.

**Weaknesses:**

1. As mentioned in the paper, "It is hard to extend GANs on large-scale datasets due to training challenges," yet there are existing works that successfully train GANs with large-scale datasets (e.g., GigaGAN with 2.7 billion training image-prompt pairs, Diffusion2GAN with 12 million training image-prompt pairs). It is important to compare these methods. While GigaGAN and Diffusion2GAN do not provide source code, the authors can still conduct comparisons using the COCO 2014 dataset, as GigaGAN provides inference results for this dataset.

GigaGAN: Scaling up GANs for Text-to-Image Synthesis (CVPR 2023)

Diffusion2GAN: Distilling Diffusion Models into Conditional GANs.

2. Why use the bottleneck layer of the UNet to compute the latent perceptual loss? Some works (e.g., DIFT, PnP, FasterDiffusion) suggest that encoder features are more important than both the encoder and bottleneck features.

DIFT: Emergent Correspondence from Image Diffusion (NeurIPS'23)

PnP: Plug-and-Play Diffusion Features for Text-Driven Image-to-Image Translation (CVPR 2023)

FasterDiffusion: Faster Diffusion: Rethinking the Role of the Encoder for Diffusion Model Inference (NeurIPS'24)

3. It’s better to provide experiments comparing with exiting SOTA one-step generative methods (e.g., SwiftBrush) to prove that YOSO enhances the inference efficiency and high-quality generation.

SwiftBrush: One-Step Text-to-Image Diffusion Model with Variational Score Distillation (CVPR'24)

4. The effectiveness of IPI should be validated on datasets like COCO 2014 using FID and CLIPscore to avoid cherry-picking. FID and CLIPscore is wildy used in one-step generation methods like SD-Turbo, InstaFlow, SwiftBrush.

**Questions:**

1. The authors did not provide information about the dataset used for the test metrics in Table 2.  Although coco2014 is mentioned in the appendix, I am not certain if it is the dataset used for these metrics.
2. The ablation study is unconvincing, as the authors used CIFAR-10 for validation. However, CIFAR-10 has a low resolution and only consists of 10 simple categories, lacking the richness of text prompts. It would be better to conduct the ablation study on the COCO 2014 and COCO 2017 datasets, similar to existing one-step text-to-image generative methods (e.g., InstaFlow, SD-Turbo, SwiftBrush).

---

> ### Author Response · Authors · 2024-11-25
> **Response (1/2)**
>
> Thank you for your time and effort in reviewing our paper.  We hereby address the concerns below.
>
> > As mentioned in the paper, "It is hard to extend GANs on large-scale datasets due to training challenges," yet there are existing works that successfully train GANs with large-scale datasets (e.g., GigaGAN with 2.7 billion training image-prompt pairs, Diffusion2GAN with 12 million training image-prompt pairs). It is important to compare these methods.
>
> We clarify that we aim to emphasize that extending pure GANs on large-scale datasets is challenging (instead of impossible), though there are existing works that successfully train GANs with large-scale datasets. In particular, despite its complicated design, GigaGAN takes 4,783 A100 days for training, which is too expensive to train. On the other hand, Diffusion2GAN is a diffusion-GAN hybrid model in a similar spirit of our YOSO. However, Diffusion2GAN requires a massive noise-image paired dataset, consuming a high cost of 15 A100 days not to mention their training cost, while our training cost via fine-tuning SD is less than 10 A800 days.
>
> > Why use the bottleneck layer of the UNet to compute the latent perceptual loss? Some works (e.g., DIFT, PnP, FasterDiffusion) suggest that encoder features are more important than both the encoder and bottleneck features.
>
> Our empirical findings show that using bottleneck features works better than using encoder features in our context. Specifically, when using encoder features, the performance of YOSO-SD-LoRA would drop from 28.33 HPS to 27.95. This may be due to the perceptual loss being more dependent on features from "deeper layers", which differs from performing other downstream image tasks.
>
> > Regarding zero-shot FID on COCO dataset.
>
> Previous work [a,b] suggests that zero-shot COCO FID may not be a good metric for evaluating modern text-to-image models. Pick-a-Pic [a] found that zero-shot COCO FID can even be negatively correlated with human preferences in certain cases. SDXL [b], which is widely known to be a more powerful generative model than SD 1.5, has a worse zero-shot COCO FID score than SD 1.5 (See Appendix F in [b]). In this work, we adopted multiple machine metrics that correlate with human preferences, including Image Reward, HPS, and AeS. These metrics actually correlate better with human preferences, as detailed in their respective papers [c,d], while our work shows the state-of-the-art performance of these metrics.
>
> Still, we are glad to provide a comparison regarding zero-shot COCO FID as follows.
>
> | Model | Latency ↓ | FID-30k ↓ | CLIP Score-30k ↑ |
> |--------|---------|--------|---------|
> | **GANs** | | | |
> |  StyleGAN-T | 0.10s | 13.90 | - |
> |  GigaGAN | 0.13s | 9.09 | - |
> | **Oringal Diffusion** | | | |
> | SD v1.5 | 2.59s | 13.45 | 0.32 |
> | **Diffusion Distillation** | | | |
> | LCM (4-step) | 0.26s | 23.62 | 0.30 |
> | Diffusion2GAN | 0.09s | 9.29 | - |
> | UFOGen | 0.09s | 12.78 | - |
> | InstaFlow-0.9B | 0.09s | 13.27 | 0.28 |
> | DMD | 0.09s | 14.93 | 0.32 |
> | SwiftBrush | 0.09s | 16.67 | 0.29 |
> | YOSO (Ours) | 0.09s | 12.35 | 0.31 |
>
> Results show that our model demonstrates competitive performance in both FID and CLIP scores when compared to modern text-to-image models and other diffusion distillation methods. This achievement is built upon our approach's superior performance in machine metrics that better represent human preferences.
>
>
> > It’s better to provide experiments comparing with exiting SOTA one-step generative methods (e.g., SwiftBrush) to prove that YOSO enhances the inference efficiency and high-quality generation.
>
> Following your suggestion, we compared SwiftBursh in terms of zero-shot FID and CLip Score in the aforementioned table. The results show that YOSO is better in both metrics. Besides, we have compared YOSO to DMD [e] in our original paper, while DMD is considered as a stronger baseline compared to SwiftBrush. In particular, both DMD and SwiftBrush are developed based on variational score distillation [f], while DMD has an additional ODE regression loss for enhancing performance.

---

> > ### Author Response · Authors · 2024-11-25
> > **Response (2/2)**
> >
> > > The authors did not provide information about the dataset used for the test metrics in Table 2. Although coco2014 is mentioned in the appendix, I am not certain if it is the dataset used for these metrics.
> >
> > We evaluate the HPS on the HPS benchmark [c], while the Image Reward Score, AeS, and clip score are evaluated on the COCO2017-5k dataset consistent with Hyper-SD.
> >
> >
> > > The ablation study is unconvincing, as the authors used CIFAR-10 for validation. However, CIFAR-10 has a low resolution and only consists of 10 simple categories, lacking the richness of text prompts.
> >
> > There might be some misunderstanding here. We conducted the ablation study on both unconditional generation on CIFAR-10 and text-to-image Synthesis on the HPS benchmark [c] and COCO2017-5k. Please note that the HPS benchmark contains a total of 3200 prompts, with 800 prompts for each of the following styles: "Animation", "Concept-art", "Painting", and "Photo". The prompts come from both COCO caption and prompts by the real user in DiffusionDB [g]. This indicates that the HPS benchmark can evaluate the text-to-image models more effectively compared to simple COCO captions.
> >
> >
> >
> > [a] Pick-a-Pic: An Open Dataset of User Preferences for Text-to-Image Generation, NeurIPS 2023.
> >
> > [b] SDXL: Improving Latent Diffusion Models for High-Resolution Image Synthesis, ICLR 2024.
> >
> > [c] Human Preference Score v2: A Solid Benchmark for Evaluating Human Preferences of Text-to-Image Synthesis, arXiv.
> >
> > [d] ImageReward: Learning and Evaluating Human Preferences for Text-to-Image Generation, NeurIPS 2023.
> >
> > [e] One-step Diffusion with Distribution Matching Distillation, CVPR 2024.
> >
> > [e] ProlificDreamer: High-Fidelity and Diverse Text-to-3D Generation with Variational Score Distillation, NeurIPS 2023.
> >
> > [f] DiffusionDB: A Large-scale Prompt Gallery Dataset for Text-to-Image Generative Models, ACL 2023.

---

> > > ### Comment · Reviewer_Awjz · 2024-11-26
> > >
> > > I appreciate the authors' response.
> > >
> > > The author has not addressed my concerns about why the bottleneck layer is used.
> > >
> > > Providing only HPS scores to prove the use of a bottleneck layer is not convincing. Almost no one has studied bottleneck layer, and the manuscript does not explain why this choice was made. This seems to be a hasty decision. The authors suggest that "This may be due to the perceptual loss being more dependent on features from deeper layers". Since the decoder layer has more "deeper layers", using the decoder layer should work better.
> > >
> > > Qualitative and quantitative ablation experiments should be provided using the encoder, bottleneck, and decoder layers, respectively, such as FID, CLIPScore, and visualization results, not just HPS scores. Additionally, would using both the encoder, bottleneck, and decoder layers to compute the latent perceptual loss be more effective?

---

> ### Author Response · Authors · 2024-11-27
>
> Thank you for your continued feedback. We appreciate the opportunity to clarify and address your concerns.
>
> **Why Use the Bottleneck Layer for Latent Perceptual Loss?**
>
> First, it is important to highlight that our contribution is proposing the latent perceptual loss (LPL) computed using features from a pre-trained latent diffusion model (DM), which contrasts with the traditional MSE loss. Our choice of using the bottleneck layer for computing LPL is motivated by its ability to provide a deeper and more **compact** feature representation. This compactness allows the perceptual loss to capture *higher-level*, abstract features, leading to improved perceptual quality in the generated images.
>
> While the decoder layers are indeed deeper in terms of network depth, their outputs are closer to the original latent space. This proximity may cause the decoder features to focus more on reconstructing *low-level* details rather than capturing the high-level perceptual aspects, which might not be as beneficial for perceptual loss purposes.
>
> **Ablation Studies on Different Layers.**
>
> It is a good suggestion. We conducted additional ablation experiments using features from the encoder, bottleneck, and decoder layers individually, as well as combining all layers for computing the LPL. Due to the limited time during the rebuttal period, we based our experiments on the LCM method for efficiency, using LoRA for fine-tuning SD 1.5 with 3k training iterations.
>
> The quantitative results are summarized below:
>
>
> | Method | HPS ↑ |Clip Score-5k ↑| NFE |
> |-------|--------|--------|--------|
> | LCM, MSE  | 15.13 | 0.242 | 1|
> | LCM, LPL-Encoder  | 19.71 | 0.278| 1|
> | LCM, LPL-Decoder  | 20.05 | 0.281 | 1|
> | LCM, LPL-All (Encoder + Bottleneck + Decoder)  | 20.13 | 0.276| 1|
> | LCM, LPL-Bottleneck  | 20.19 | 0.281 | 1|
>
> | Method | HPS ↑ |Clip Score-5k ↑| NFE |
> |-------|--------|--------|--------|
> | LCM, MSE  | 20.56 | 0.293 | 2|
> | LCM, LPL-Encoder  | 23.66 | 0.303| 2|
> | LCM, LPL-Decoder  | 23.35 | 0.298| 2|
> | LCM, LPL-All (Encoder + Bottleneck + Decoder)  | 24.00 | 0.302| 2|
> | LCM, LPL-Bottleneck  | 24.26 | 0.305| 2|
>
> We also include visual comparisons among  variants in Figure 11 located in the Appendix.
>
> From both the quantitative results and qualitative comparisons, we observe:
>
> - Using bottleneck features for computing LPL achieves the best performance, outperforming other variants in terms of HPS and CLIP Scores.
> - LPL consistently outperforms MSE, regardless of which feature layer is used, demonstrating the effectiveness of the latent perceptual loss as a whole.
> - While combining all layers (Encoder + Bottleneck + Decoder) does improve over using individual encoder or decoder layers, it does not surpass the performance of using the bottleneck layer alone. This suggests that the bottleneck layer captures the most relevant features for the perceptual loss in our context.
>
> **Regarding Evaluation Metrics.** We would like to reiterate that zero-shot FID is not a reliable metric for evaluating text-to-image models, as discussed in prior works [a, b]. CLIP Score is a more appropriate metric in this context, and we have provided the additional CLIP Scores as you recommended.
>
> [a] Pick-a-Pic: An Open Dataset of User Preferences for Text-to-Image Generation, NeurIPS 2023.
>
> [b] SDXL: Improving Latent Diffusion Models for High-Resolution Image Synthesis, ICLR 2024.

---

> > ### Comment · Reviewer_Awjz · 2024-11-29
> >
> > I have a different perspective on the view that the output of decoder layers is closer to the original latent space. In SD, the decoder consists of multiple layers at different scales. Only the output of the final layer is closer to the original latent space, while the outputs of the other layers in the decoder are similar to the bottleneck's output and even contain richer information (see Figure 2 in [A] and Figure 3 in [B]). It would be more appropriate to use the outputs of different-scale layers in the decoder to compute the perceptual loss.
> >
> > [A] DeepCache: Accelerating Diffusion Models for Free.\
> > [B] Plug-and-Play Diffusion Features for Text-Driven Image-to-Image Translation.
> >
> > Which scale layer's output from the decoder is used in the new table? The output of the final layer in the decoder is closer to the latent code and is not suitable for computing the perceptual loss.
> >
> > Most of the existing one-step methods [C, D, E] report the FID metric, which has reference value.
> >
> > [C] SwiftBrush : One-Step Text-to-Image Diffusion Model with Variational Score Distillation\
> > [D] Adversarial Diffusion Distillation\
> > [E] Distilling Diffusion Models into Conditional GANs

---

> ### Author Response · Authors · 2024-11-30
>
> Thank you for your valuable feedback. We are glad to provide further clarification.
>
> > I have a different perspective on the view that the output of decoder layers is closer to the original latent space. In SD, the decoder consists of multiple layers at different scales. Only the output of the final layer is closer to the original latent space, while the outputs of the other layers in the decoder are similar to the bottleneck's output and even contain richer information (see Figure 2 in [A] and Figure 3 in [B]). It would be more appropriate to use the outputs of different-scale layers in the decoder to compute the perceptual loss. Which scale layer's output from the decoder is used in the new table? The output of the final layer in the decoder is closer to the latent code and is not suitable for computing the perceptual loss.
>
> **Clarification on Our View of Outputs of Decoder Layers** We clarify that our statement that "the outputs of decoder layers are closer to the original latent space" is made in comparison to the encoder layers or the bottleneck layer. Intuitively, as we progress deeper into the decoder, the feature representations tend to more closely approximate the original latent space. For instance, as observed in Figure 3 of [B], the features from decoder layer 4 are significantly closer to the original latent space compared to those from layer 1. Similarly, the features from layer 7 are closer to the original latent space than those from layer 4.
>
>
> **Decoder Layers Used in Our Experiments** We compute the latent perceptual loss using all decoder layers except the last one.
>
> We agree that decoder layers may contain richer details, but too many details remaining might make the network focus more on reconstructing low-level details when computing the perceptual loss, which could be detrimental to the calculation of perceptual loss.
>
> To investigate this further,  we conduct additional experiments in using different-scale decoder layers for computing the LPL, similar to the setting in [B]. The results are presented in the tables below.
>
> | Method | HPS ↑ |FID-5k ↓ | Clip Score-5k ↑| NFE |
> |-------|--------|--------|--------| --------|
> | LCM, MSE  | 15.13 | 87.47 | 0.242 |  1|
> | LCM, LPL-Encoder  | 19.71 | 37.90| 0.278| 1|
> | LCM, LPL-Decoder (layers = 1)  | 20.15 | 39.16| 0.281 | 1|
> | LCM, LPL-Decoder (layers = 4)  | 19.40 | 42.45| 0.277 | 1|
> | LCM, LPL-Decoder (layers = 4-8)  | 18.71 | 43.71| 0.273 | 1|
> | LCM, LPL-Decoder (layers = 4-10)  | 19.45 | 43.66| 0.279 | 1|
> | LCM, LPL-Decoder  | 20.05 | 40.51| 0.281 | 1|
> | LCM, LPL-All (Encoder + Bottleneck + Decoder)  | 20.13 | 37.30| 0.276| 1|
> | LCM, LPL-Bottleneck  | 20.19 | 37.51| 0.281 | 1|
>
> | Method | HPS ↑ |FID-5k ↓ | Clip Score-5k ↑| NFE |
> |-------|--------|--------|--------|--------|
> | LCM, MSE  | 20.56 | 30.69| 0.293 | 2|
> | LCM, LPL-Encoder  | 23.66 | 26.33| 0.303| 2|
> | LCM, LPL-Decoder (layers = 1)  | 24.35 | 25.06| 0.305 | 2|
> | LCM, LPL-Decoder (layers = 4)  | 23.40 | 26.82| 0.302 | 2|
> | LCM, LPL-Decoder (layers = 4-8)  | 22.83 | 27.08| 0.299 | 2|
> | LCM, LPL-Decoder (layers = 4-10)  | 23.26 | 27.35| 0.300 | 2|
> | LCM, LPL-Decoder  | 23.35 | 27.49| 0.298| 2|
> | LCM, LPL-All (Encoder + Bottleneck + Decoder)  | 24.00 | 25.87| 0.302| 2|
> | LCM, LPL-Bottleneck  | 24.26 | 24.21| 0.305| 2|
>
> It can be observed that using decoder layers=1 clearly performs the best among the different decoder layer configurations. This is different from the observation in [B], where they found layers=4-11 performs best in their context.
>
> This is because [B] aims to generate images that are similar but not identical to the reference image by replacing the outputs/features of certain layers in the diffusion network. This direct replacement operation can only leverage the outputs/features themselves, making it necessary to preserve as much detail as possible (without leaking appearance) to retain the semantics of the reference image.
>
> In our case, the LPL optimizes $||f(x)-f(\text{target})||_2^2$, where the student network is trained via backpropagation through $f(x)$. This allows it to not only leverage the features themselves but also utilize the gradient information of the perceptual network $f$ (i.e., the pre-trained diffusion). As a result, the layer selection that is suitable for [B] might lead the perceptual loss to overly focus on low-level details in our case.
>
> More importantly, **regardless of which layers are used to compute LPL, it is significantly better than MSE**. This highlights the effectiveness of our proposed latent perceptual loss.

---

> > ### Author Response · Authors · 2024-12-02
> >
> > Dear Reviewer Awjz,
> >
> > We sincerely appreciate your thorough review and valuable feedback. We fully understand you might be quite busy. However, as the discussion deadline is approaching, would you mind checking our follow-up additional response? We would like to know if our recent efforts have addressed your remaining concerns. We also welcome any further comments or discussions you may have.
> >
> > Thank you once again for your time and effort in reviewing our paper.
> >
> > Many thanks,
> >
> > Authors

---

> > ### Author Response · Authors · 2024-12-04
> >
> > As the discussion period is nearing its end, we kindly request Reviewer Awjz to check our additional response. If there are any further questions or matters to discuss, please don’t hesitate to let us know.
> >
> > Thanks again for your time and effort in reviewing our paper.

---

### Official Review · Reviewer_SYJU · 2024-10-30

**Soundness:** 3
**Presentation:** 2
**Contribution:** 2
**Rating:** 6
**Confidence:** 4

**Summary:**

This paper, "You Only Sample Once: Taming One-Step Text-to-Image Synthesis by Self-Cooperative Diffusion GANs (YOSO)", proposes a novel generative method that merges diffusion models (DMs) with GANs to perform efficient, high-quality one-step text-to-image synthesis. YOSO leverages a self-cooperative learning framework, where a generator smooths adversarial divergence by referencing self-generated denoised samples, combined with a latent perceptual loss and a decoupled scheduler to stabilize training. The paper positions YOSO as an efficient alternative to multi-step latent diffusion methods (LDMs) with discriminator-based losses, aiming to achieve comparable quality while reducing computational overhead. Experimental results demonstrate competitive performance with state-of-the-art methods, though the model would benefit from clearer differentiation from similar single-step latent diffusion approaches.

**Strengths:**

1. Creative Approach to Stability in One-Step Synthesis: YOSO’s self-cooperative learning framework is an inventive attempt to stabilize adversarial training by smoothing divergence via self-generated data, a strategy distinct from standard GAN-DM hybrids. This approach may address some instability challenges faced by prior one-step latent diffusion models.
2. Efficient Resource Utilization: YOSO’s compatibility with LoRA allows it to produce high-resolution images without the significant computational burden typically associated with GAN-based models, making it adaptable for resource-constrained settings.

**Weaknesses:**

1. Clarity of Key Contribution: This paper lacks clarity on whether YOSO is a new model architecture that can be trained from scratch or a step distillation methodology that should be applied on pre-trained models. Further elaboration, possibly with additional diagrams, would make the paper more accessible and clear.

2. Experimental Scope and Robustness: This paper introduces YOSO both as a new model and a novel distillation techniques for pre-trained models. However, the former part was only evaluated on the CIFAR-10 dataset, additional benchmarks are necessary to substantiate its effectiveness across more diverse and complex datasets.

2. Novelty in Methodology: Besides self-cooperative learning, the adoption of decoupled scheduling, perceptual and consistency loss isn't novel to the field, and while the rationale is sound, these sections could benefit from more detailed comparison with previous works and how it is different from them.

**Questions:**

Please refer to the weakness section

---

> ### Author Response · Authors · 2024-11-25
>
> Thank you for your time and effort in reviewing our paper. We very much appreciate your acknowledgment of our proposed self-cooperative learning and the efficiency of our YOSO. We hereby address the concerns below.
>
>
> > Clarity of Key Contribution: This paper lacks clarity on whether YOSO is a new model architecture that can be trained from scratch or a step distillation methodology that should be applied on pre-trained models. Further elaboration, possibly with additional diagrams, would make the paper more accessible and clear.
>
> As clearly stated in the abstract, YOSO can support both training from scratch and finetuning pre-trained diffusion models, since the loss for training YOSO does not depend on a pre-trained diffusion model. We have included the algorithm description of training YOSO from scratch in Algorithm 1 which is located in Appendix D in our original manuscript.
>
> > Experimental Scope and Robustness: This paper introduces YOSO both as a new model and a novel distillation techniques for pre-trained models. However, the former part was only evaluated on the CIFAR-10 dataset, additional benchmarks are necessary to substantiate its effectiveness across more diverse and complex datasets.
>
>
> To further demonstrate that YOSO can serve as a new model trained from scratch, we conducted additional experiments on ImageNet-64 that adopt EDM architecture on both generator and discriminator for training YOSO from scratch. The results are shown in the following table:
>
> **Table A**: Results obtained by training from scratch
> | Method | NFE | FID (↓) |
> |--------|---------------|---------|
> | BigGAN-deep  | 1 | 4.06 |
> | StyleGAN-XL | 1 | 1.52 |
> | ADM  | 250 | 2.07 |
> | EDM | 511 | 1.36 |
> | CT | 1 | 13.0 |
> | iCT  | 1 | 3.25 |
> | YOSO （Ours）  | 1 | 2.65 |
>
> It can be seen that YOSO has clearly better performance than CT and iCT, and achieves comparable performance to state-of-the-art diffusion models and GANs.
>
> > Novelty in Methodology: Besides self-cooperative learning, the adoption of decoupled scheduling, perceptual and consistency loss isn't novel to the field, and while the rationale is sound, these sections could benefit from more detailed comparison with previous works and how it is different from them.
>
> Our proposed decoupled scheduler integrates the GAN loss and diffusion distillation loss (consistency loss) in a simple and effective way, which ensures that we just train one stage. In contrast, although some works [a, b] have combined GAN loss with diffusion distillation loss, without our decoupled scheduler, they have to train the model in multiple stages with the scheduler coupled between GAN loss and distillation loss. Moreover, the proposed Latent Perceptual Loss provides a method to efficiently compute perceptual loss in ***latent*** space using pre-trained diffusion models. In contrast, previous work [c] requires decoding latent through a VAE decoder and then computing LPIPS loss in data space, which is computationally expensive.
>
> [a] SDXL-Lightning: Progressive Adversarial Diffusion Distillation, arXiv.
>
> [b] Hyper-SD: Trajectory Segmented Consistency Model for Efficient Image Synthesis, NeurIPS 2024.
>
> [c] InstaFlow: One Step is Enough for High-Quality Diffusion-Based Text-to-Image Generation, ICLR 2024.

---

> > ### Comment · Reviewer_SYJU · 2024-11-27
> >
> > Dear authors,
> >
> > Thanks for the additional experiments and follow up explanations, I have no further questions and raised my score.

---

> > > ### Author Response · Authors · 2024-11-27
> > >
> > > We thank you for acknowledging our work and for raising the score. Thanks again for your time and effort in reviewing our work.

---

### Official Review · Reviewer_RjVy · 2024-11-02

**Soundness:** 2
**Presentation:** 2
**Contribution:** 2
**Rating:** 5
**Confidence:** 4

**Summary:**

This paper introduces YOSO, a novel generative model designed for rapid, scalable, and high-fidelity one-step image synthesis with high training stability and mode coverage. Specifically, the paper smooths the adversarial divergence by the denoising generator itself, performing self-cooperative learning. This method can serve as a one-step generation model training from scratch with competitive performance. Moreover, the authors extend YOSO to one-step text-to-image generation based on pre-trained models by several effective training techniques (i.e., latent perceptual loss and latent discriminator for efficient training along with the latent DMs; the informative prior initialization (IPI), and the quick adaption stage for fixing the flawed noise scheduler). Experimental results show that YOSO achieves state-of-the-art one-step generation performance even with Low-Rank Adaptation (LoRA) fine-tuning.

**Strengths:**

1. The article is well written and easy to understand.
2. The visualization results look very good.

**Weaknesses:**

1. There is a lack of important references in the Background, such as [1,2,3], which are all about combining Diffusion models and GANs.  The differences between the proposed method and these methods [1,2,3] need to be emphasized and discussed.
2. In the experimental part, it is also necessary to compare with these methods [1,2,3].
3. There is no comparison with the current SOTA methods in terms of model complexity (such as the number of network parameters and training inference time).
4. In the TEXT-TO-IMAGE GENERATION experiment, there is no comparison with the GAN-based methods, such as [4]. It is also worth discussing whether the proposed method is better or more efficient than the GAN-based methods.

[1]Wang, Zhendong, Huangjie Zheng, Pengcheng He, Weizhu Chen, and Mingyuan Zhou. "Diffusion-gan: Training gans with diffusion." arXiv preprint arXiv:2206.02262 (2022).
[2]Kang, Minguk, Richard Zhang, Connelly Barnes, Sylvain Paris, Suha Kwak, Jaesik Park, Eli Shechtman, Jun-Yan Zhu, and Taesung Park. "Distilling Diffusion Models into Conditional GANs." ECCV (2024).
[3]Xu, Yanwu, Yang Zhao, Zhisheng Xiao, and Tingbo Hou. "Ufogen: You forward once large scale text-to-image generation via diffusion gans." In Proceedings of the IEEE/CVF Conference on Computer Vision and Pattern Recognition, pp. 8196-8206. 2024.
[4]Kang, Minguk, Jun-Yan Zhu, Richard Zhang, Jaesik Park, Eli Shechtman, Sylvain Paris, and Taesung Park. "Scaling up gans for text-to-image synthesis." In Proceedings of the IEEE/CVF Conference on Computer Vision and Pattern Recognition, pp. 10124-10134. 2023.

**Questions:**

See Weaknesses.

---

> ### Author Response · Authors · 2024-11-25
>
> Thank you for your time and effort in reviewing our paper.  We hereby address the concerns below.
>
>
> > There is a lack of important references in the Background, such as [1,2,3], which are all about combining Diffusion models and GANs. The differences between the proposed method and these methods [1,2,3] need to be emphasized and discussed.
>
> Thanks for mentioning these excellent papers. We have discussed UFOGen [3] in our original manuscript, and we provide a discussion of [1,2] as follows. Similar to UFOGen [3], the method proposed in [1] injects noise into samples to stabilize adversarial training. This leads to subpar one-step generation learning efficiency. On the other hand, diffusion2GAN [2] relies on using diffusion models to construct numerous noise-image pairs (3M pairs in finetuning SD) for organizing regression loss to stabilize training, which is computationally expensive. Specifically, constructing 3M noise-image pairs using SD 1.5 requires 15 A100 days according to the data in InstaFlow, while our computational cost for fine-tuning SD using LoRA is less than 10 A800 days. This highlights the efficiency of our proposed method.
>
>
> > In the experimental part, it is also necessary to compare with these methods [1,2,3].
>
> In the original paper, we compared UFOgen [3] on CIFAR-10, with results shown in Table 3 of the paper. Below we provide an experimental comparison with [1, 2] on CIFAR-10:
>
> | Model | FID↓ |
> |-------|--------|
> | Diffusion-GAN [1]  | 3.19 |
> | Diffusion2GAN [2]  | 3.16 |
> | YOSO (Ours) | 1.81 |
>
> It can be seen that our approach achieves a significantly lower FID score of 1.81, representing a 43% improvement over the next best method, Diffusion2GAN [2] at 3.16.
>
> > There is no comparison with the current SOTA methods in terms of model complexity (such as the number of network parameters and training inference time).
>
> In comparison with diffusion distillation methods, we consistently maintain the same backbone for the student model, ensuring that all methods have identical parameter counts and single-step inference costs. Moreover, our proposed YOSO just requires less than 10 A800 GPU days to fine-tune SD 1.5, while InstaFlow requires 199 A100 GPU days to fine-tune SD 1.5 and most other works (e.g., SD-Turbo [a], PeRFlow [b], and Hyper-SD [c]) do not report their training cost.
>
> [a] Adversarial Diffusion Distillation, ECCV 2024.
>
> [b] PeRFlow: Piecewise Rectified Flow as Universal Plug-and-Play Accelerator, NeurIPS 2024.
>
> [c] Hyper-SD: Trajectory Segmented Consistency Model for Efficient Image Synthesis, NeurIPS 2024.
>
>
> > In the TEXT-TO-IMAGE GENERATION experiment, there is no comparison with the GAN-based methods, such as [4]. It is also worth discussing whether the proposed method is better or more efficient than the GAN-based methods.
>
> Our approach has a significant advantage in training costs compared to pure GANs, since we can efficiently fine-tune the pre-trained diffusion models. Specifically, while GigaGAN [4] requires 4,783 A100-days for training, we need less than 10 A800-days to fine-tune a pre-trained diffusion model (SD-v1.5 and PixArt-$\alpha$).

---

> > ### Comment · Reviewer_RjVy · 2024-11-26
> > **The response did not address my concerns.**
> >
> > Thank you for your answers, but they do not address my concerns.
> >
> > 1. It is not enough to compare only on CIFAR-10, which has only 32X32 resolution. Experimental comparisons should be conducted on CelebA (64X64), LSUN-Bedroom (256X256), LSUN-Church (256X256), and FFHQ (1024X1024) datasets with the three methods I pointed out [1,2,3].
> > 2. Regarding model complexity, the author should compare it like Table 1 and Table 2 in the UFOGen paper.
> > 3. A fair comparison should be conducted with GigaGAN under the same experimental conditions.

---

> > > ### Author Response · Authors · 2024-11-27
> > > **Follow-up Response (1/2)**
> > >
> > > Thank you for your continued feedback. We appreciate the opportunity to clarify and address your concerns.
> > >
> > > > It is not enough to compare only on CIFAR-10, which has only 32X32 resolution. Experimental comparisons should be conducted on CelebA (64X64), LSUN-Bedroom (256X256), LSUN-Church (256X256), and FFHQ (1024X1024) datasets with the three methods I pointed out [1,2,3].
> > >
> > > **Additional Comparisons on Zero-Shot COCO FID.** We have extended our comparisons to include methods from [1] and [3] on zero-shot COCO FID, as detailed in Point 4 of our general response. We attach the table below for ease of reference.
> > >
> > > **Table A**: Comparison of Zero-shot COCO FID and Clip score.
> > > | Model | Latency ↓ | Params | FID-30k ↓ | CLIP Score-30k ↑ |
> > > |--------|---------|---------|--------|---------|
> > > | **GANs** | | | | |
> > > |  StyleGAN-T | 0.10s | 1B | 13.90 | - |
> > > |  GigaGAN | 0.13s | 1B | 9.09 | - |
> > > | **Oringal Diffusion** | | | | |
> > > | SD v1.5 | 2.59s | 0.9B | 13.45 | 0.32 |
> > > | **Diffusion Distillation** | | | | |
> > > | LCM (4-step) | 0.26s | 0.9B | 23.62 | 0.30 |
> > > | Diffusion2GAN | 0.09s | 0.9B | 9.29 | - |
> > > | UFOGen | 0.09s | 0.9B | 12.78 | - |
> > > | InstaFlow-0.9B | 0.09s | 0.9B | 13.27 | 0.28 |
> > > | DMD | 0.09s | 0.9B | 14.93 | 0.32 |
> > > | SwiftBrush | 0.09s | 0.9B | 16.67 | 0.29 |
> > > | YOSO (Ours) | 0.09s | 0.9B | 12.35 | 0.31 |
> > > | YOSO （Ours, retrained, full fine-tuned, selected by FID）  | 0.09s | 0.9B | 9.61 | 0.31 |
> > >
> > >
> > > From Table A, we observe that our method demonstrates competitive performance compared to existing state-of-the-art models in terms of zero-shot FID and CLIP Score, further validating its effectiveness.  Please note that this achievement is built upon our approach's superior performance in machine metrics that better represent human preferences.
> > >
> > > **Comparison with SIDDMs [b] on ImageNet-64.** We have also compared our method with SIDDMs [b] on the ImageNet-64 dataset. It is noteworthy that SIDDMs is the predecessor of UFOGen [3], with UFOGen extending SIDDMs to text-to-image generation.
> > >
> > > Evaluating methods on CIFAR-10 and ImageNet-64 is a widely accepted practice in the community [c,d]. Our inclusion of ImageNet-64 aligns with this standard, providing a meaningful comparison. Note that ImageNet has 1000 categories, each containing
> > > thousands of images and distinguishing itself as a large-scale and challenging dataset with
> > > diversity.
> > >
> > > **Table B**: Results obtained by training from scratch on ImageNet-64.
> > > | Method | NFE | FID (↓) |
> > > |--------|---------------|---------|
> > > | BigGAN-deep  | 1 | 4.06 |
> > > | StyleGAN-XL | 1 | 1.52 |
> > > | ADM  | 250 | 2.07 |
> > > | EDM | 511 | 1.36 |
> > > | CT | 1 | 13.0 |
> > > | iCT  | 1 | 3.25 |
> > > | SIDDMs | 4 | 3.13 |
> > > | YOSO （Ours）  | 1 | 2.65 |
> > >
> > >
> > > As shown in Table B, our method achieves a better FID score compared to SIDDMs while using fewer NFEs, indicating both efficiency and improved performance.
> > >
> > >
> > > **Regarding the Zero-shot FID metric.** While we provide the comparison on zero-shot COCO FID, it is important to note that zero-shot COCO FID is not a good metric for evaluating text-to-image models. Specifically, Pick-a-Pic [a] found that zero-shot COCO FID can even be negatively correlated with human preferences in certain cases. SDXL [b], which is widely known to be a more powerful generative model than SD 1.5, has a worse zero-shot COCO FID score than SD 1.5 (See Appendix F in [b]). In this work, we adopted multiple machine metrics that correlate with human preferences, including Image Reward, HPS, and AeS. These metrics actually correlate better with human preferences, as detailed in their respective papers [c,d], while our work shows the state-of-the-art performance of these metrics.
> > >
> > >
> > > **Consistency with Prior Works.** Diffusion2GAN [2], one of the methods you pointed out, evaluated their approach on CIFAR-10 and text-to-image synthesis in their original paper, which aligns with our initial evaluation strategy. UFOGen [1] evaluated their approach on text-to-image synthesis in their paper. Therefore, our comparative analysis remains consistent with the evaluation settings commonly adopted in prior works. Besides, it is also important to note that Diffusion2GAN should be considered as a concurrent work accordint to the reviewer guide of ICLR (https://iclr.cc/Conferences/2025/ReviewerGuide), since it is appeared in ECCV 24 which was published within the last four months of ICLR deadline.
> > >
> > > **Alignment with Community Practices.** Existing works on text-to-image diffusion acceleration often focus on specific datasets or evaluation settings. Some studies conducted experiments exclusively on text-to-image tasks [3,f,g,h,i], while others included additional experiments on CIFAR-10 or ImageNet-64 [1,e]. By providing experiments on both CIFAR-10 and ImageNet-64, as well as text-to-image generation, we believe our work aligns well with the practices in the research community.

---

> ### Author Response · Authors · 2024-11-27
> **Follow-up Response (2/2)**
>
> > Regarding model complexity, the author should compare it like Table 1 and Table 2 in the UFOGen paper.
>
> Please refer to Table A above. It can be seen that YOSO is one of the most efficient models.
>
>
> > A fair comparison should be conducted with GigaGAN under the same experimental conditions.
>
> We provide a comparison of zero-shot COCO FID with GigaGAN. Please refer to Table A above.
>
> **References.**
>
> [a] Pick-a-Pic: An Open Dataset of User Preferences for Text-to-Image Generation, NeurIPS 2023.
>
>
> [b] Semi-Implicit Denoising Diffusion Models (SIDDMs), NeurIPS 2023.
>
> [c] Maximum Likelihood Training of Score-Based Diffusion Models, NeurIPS 2021.
>
> [d] Improved Techniques for Training Consistency Models, ICLR 2024.
>
> [e] One-step Diffusion with Distribution Matching Distillation, CVPR 2024.
>
> [f] SwiftBrush: One-Step Text-to-Image Diffusion Model with Variational Score Distillation, CVPR 2024.
>
> [g] InstaFlow: One Step is Enough for High-Quality Diffusion-Based Text-to-Image Generation, ICLR 2024.
>
> [h] Hyper-SD: Trajectory Segmented Consistency Model for Efficient Image Synthesis, NeurIPS 2024.
>
> [i] Adversarial Diffusion Distillation, ECCV 2024.
>
> [j] SDXL: Improving Latent Diffusion Models for High-Resolution Image Synthesis, ICLR 2024.
>
> [k] Human Preference Score v2: A Solid Benchmark for Evaluating Human Preferences of Text-to-Image Synthesis, arXiv.
>
> [l] ImageReward: Learning and Evaluating Human Preferences for Text-to-Image Generation, NeurIPS 2023.

---

> > ### Comment · Reviewer_RjVy · 2024-11-27
> > **No high-resolution (1024X1024) comparison results, and performance is worse than GigaGAN**
> >
> > Thank you for your response. However, I would appreciate it if the authors could provide comparison results with state-of-the-art (SOTA) methods on high-resolution benchmarks such as FFHQ (1024×1024). Additionally, I would be interested in seeing text-conditioned 128→1024 super-resolution results on random 10K LAION samples, similar to what was presented in Table 4 of the GigaGAN paper.
> >
> > In addition, Table A shows that the proposed method does not outperform GigaGAN (or Diffusion2GAN) on FID-30K.

---

> > > ### Author Response · Authors · 2024-11-30
> > >
> > > We appreciate your follow-up comments and are happy to address them in more detail.
> > >
> > > > I would appreciate it if the authors could provide comparison results with SOTA methods on high-resolution benchmarks such as FFHQ (1024×1024).
> > >
> > > Thank you for this suggestion. We are glad to provide additional experiments in comparing methods from [1,2,3] on FFHQ-1024 below.
> > >
> > > **Experiment Setting**: Due to the time limitation, we conducted comparative experiments between methods by fine-tuning pre-trained diffusion models. To the best of our knowledge, the latest publicly available pre-trained diffusion model on FFHQ-1024 is NCSNpp [a], a score-based model in data space. We found that generating 50k samples with NCSNpp to compute FID would require tens of A800 GPU days, which is too computationally expensive. Therefore, we trained a lightweight diffusion model in the latent space of DC-AE [b] for efficiency. Specifically, following DC-AE, we adopted DiT-S as the model backbone, which is a lightweight model with 33M parameters. For a fair comparison, we initialized the generator for YOSO as well as the methods from [1,2,3] using the pre-trained diffusion model.
> > >
> > > **Table A**: Results on FFHQ-1024.
> > > | Method | NFE | FID ↓ |
> > > |--------|---------------|---------|
> > > | Diffusion (Results from DC-AE)  | 250 | 13.65|
> > > | Teacher Diffusion (Our reproduced)  | 250 | 13.88 |
> > > | **Diffusion Distillation** | | | | |
> > > | Diffusion-GAN [1]  | 1 | 22.16 |
> > > | Diffusion2GAN [2] | 1 | 18.01 |
> > > | UFOGen [3] | 1 | 24.37 |
> > > | YOSO （Ours）  | 1 | 16.23 |
> > >
> > > As shown in Table A, YOSO achieves better performance on FFHQ-1024 compared to other distillation methods.
> > >
> > > [a] Improved Techniques for Training Score-Based Generative Models, Neurips 2020.
> > >
> > > [b] Deep Compression Autoencoder for Efficient High-Resolution Diffusion Models, ICLR 2025 Submission.
> > >
> > > > Additionally, I would be interested in seeing text-conditioned 128→1024 super-resolution results on random 10K LAION samples, similar to what was presented in Table 4 of the GigaGAN paper.
> > >
> > > Good suggestion. We are glad to provide additional experiments on 128→512 upscaling because the SD upscaler was specifically trained for the 128→512 super-resolution task.  We initialized YOSO with the SD upscaler. We report the results below. We evaluated several metrics, including FID, PSNR, LPIPS, and CLIP Score. The FID is computed between 10k ground truth samples and 10k generated samples, measuring the fidelity of the images. The CLIP Score assesses the text-image alignment, while PSNR and LPIPS measure the consistency between the generated samples and the ground truth samples.
> > >
> > > **Table B**: Results on Super-Resolution (128 -> 512).
> > > | Method | NFE | FID ↓ |  PSNR ↑ | LPIPS ↓ | Clip Score ↑|
> > > |--------|---------------|---------|---------| ---------|---------|
> > > | SD upscaler   | 50 | 5.81| 19.7 | 0.29 | 0.32 |
> > > | YOSO （Ours）  | 1 | 6.15 | 20.1 | 0.27 | 0.32 |
> > >
> > > As we can observe, although YOSO achieved an approximately 50x speedup compared to SD upscaler, it obtained better consistency measured by PSNR and LPIPS while only showing slightly worse FID scores. This demonstrates YOSO's promising potential in accelerating pretrained super-resolution diffusion models.
> > >
> > > > In addition, Table A shows that the proposed method does not outperform GigaGAN (or Diffusion2GAN) on FID-30K.
> > >
> > > This does not mean that our model is inferior, since zero-shot COCO FID is not the gold standard for evaluating modern text-to-image models. Notably, even SD (cfg=7.5) does not surpass Diffusion2GAN and GigaGAN in terms of zero-shot COCO FID. Although SD can indeed achieve a lower zero-shot COCO FID-30k of 8.78 by setting CFG =3, however, this comes at the cost of significantly reduced visual quality, as studied in Pick-a-Pic [c].
> > >
> > > Additionally, we found that continuing to train the YOSO (FID = 9.61) checkpoint for 10k iterations can achieve a lower zero-shot COCO FID. The updated results are presented below.
> > >
> > > **Table C**: Comparsion of Zero-shot COCO FID-30k.
> > > | Model | Latency ↓ | Params | FID-30k ↓ |
> > > |--------|---------|---------|--------|
> > > | **GANs** | | | |
> > > |  StyleGAN-T | 0.10s | 1B | 13.90 |
> > > |  GigaGAN | 0.13s | 1B | 9.09 |
> > > | **Original Diffusion** | | | |
> > > | SD v1.5 (CFG=7.5) | 2.59s | 0.9B | 13.45 |
> > > | SD v1.5 (CFG=3) | 2.59s | 0.9B | 8.78 |
> > > | **Diffusion Distillation** | | | |
> > > | Diffusion2GAN | 0.09s | 0.9B | 9.29 |
> > > | YOSO (Ours) | 0.09s | 0.9B | 12.35 |
> > > | YOSO（Ours, retrained, full fine-tuned, selected by FID）  | 0.09s | 0.9B | 9.61 |
> > > | YOSO（Ours, longer training）  | 0.09s | 0.9B | 8.90 |
> > >
> > > This indicates that YOSO has the capability to achieve a lower zero-shot COCO FID compared to GigaGAN. However, we would like to highlight that our primary goal in this work is to develop a one-step text-to-image model with better text-image alignment and visual quality, rather than targeting the SOTA zero-shot COCO FID score. Hope this makes things clearer.
> > >
> > > [c] Pick-a-Pic: An Open Dataset of User Preferences for Text-to-Image Generation, NeurIPS 2023.

---

> > > > ### Author Response · Authors · 2024-12-02
> > > >
> > > > Dear Reviewer RjVy,
> > > >
> > > > We sincerely appreciate your thorough review and valuable feedback. We fully understand you might be quite busy. However, as the discussion deadline is approaching, would you mind checking our follow-up additional experiments and response? We would like to know if our recent efforts have addressed your remaining concerns. We also welcome any further comments or discussions you may have.
> > > >
> > > > Thank you once again for your time and effort in reviewing our paper.
> > > >
> > > > Many thanks,
> > > >
> > > > Authors

---

> > > > ### Author Response · Authors · 2024-12-04
> > > >
> > > > As the discussion period is nearing its end, we kindly request Reviewer RjVy to check our additional response. If there are any further questions or matters to discuss, please don’t hesitate to let us know.
> > > >
> > > > Thanks again for your time and effort in reviewing our paper.

---

### Official Review · Reviewer_tJNk · 2024-11-03

**Soundness:** 3
**Presentation:** 2
**Contribution:** 3
**Rating:** 6
**Confidence:** 4

**Summary:**

This paper combines diffusion models and GANs to propose the YOSO framework for one-step image generation. It employs various training strategies to enhance training stability and has been validated on both unconditional generation and text-to-image tasks.

**Strengths:**

- In terms of qualitative metrics, YOSO demonstrates certain advantages across various tasks and metrics, and it is capable of transferring to higher resolutions, image editing, and conditional generation tasks.

- The simple and effective training strategies proposed during practical implementation have certain reference significance for generative tasks.

- The paper shows many analyses and easy to follow.

**Weaknesses:**

- The presentation of the visualization results in this paper is somewhat insufficient, including a comparison with YOSO-PixArt-alpha and other Few-Step models, such as PixArt-delta, and it might be worth attempting to compare with the original models as well.

- The clarity of the narrative in this paper needs improvement; currently, the narrative logic resembles an experimental report. When encountering issues in a new task (unconditional and text-to-image generation / finetuning) or a new model (U-Net or DiT based), a specific strategy is proposed to address the problem. This approach may not be conducive to the coherence expected in a complete paper.

- Regarding writing:
  - The abbreviation "NEF" is not explained.
  - Should the steps for Hyper-SD-LoRA and YOSO-LoRA in Table 2 be equal to 1?
  - Figure 5 does not specify the comparative configuration for model inference—does it keep the prompt unchanged while changing the same seed?
  - The reference formatting in the paper should be kept consistent as possible.

**Questions:**

- Why were only SD-LoRA and PixArt-Full experiments conducted in the experimental setup, without including SD-Full and PixArt-LoRA related experiments?

- I am curious about how the zero-shot one-step 1024 resolution is effective. As far as I know, 1024 has a different ar_size compared to 512, and the posemb scale is also different. Wouldn't these factors significantly impact the weight distribution of the 1024 model?

- Without retraining, will increasing the sampling steps, similar to LCM, improve the results? If further enhancements are desired for better generated image quality, what would be the subsequent optimization directions based on the proposed approach?

---

> ### Author Response · Authors · 2024-11-25
>
> Thank you for your time and effort in reviewing our paper. We very much appreciate your acknowledgement of our proposed training strategies and the advantages over qualitative metrics. We hereby address the concerns below.
>
>
> > The presentation of the visualization results in this paper is somewhat insufficient, including a comparison with YOSO-PixArt-alpha and other Few-Step models, such as PixArt-delta, and it might be worth attempting to compare with the original models as well.
>
> Thanks for your advice. We have included the visualization comparison to PixArt-$\alpha$ (original teacher model) and PixArt-$\delta$ on 1024 resolution in Figure 9 located in the appendix. As can be observed, our method produces significantly better image quality compared to PixArt-$\delta$ and achieves comparable results to the multi-step teacher model.
>
>
>
> > The abbreviation "NEF" is not explained.
>
> The "NFE" denotes Number of Function Evaluations. We have explained this abbreviation in our revision.
>
> > Should the steps for Hyper-SD-LoRA and YOSO-LoRA in Table 2 be equal to 1?
>
> Not really. Table 2 evaluates the performance of Hyper-SD-LoRA and YOSO-LoRA in both 1-step and 4-step generation.
>
> > Figure 5 does not specify the comparative configuration for model inference—does it keep the prompt unchanged while changing the same seed?
>
> Yes. We keep the prompt unchanged while changing the random seed. The random seed is the same for each image among models.
>
> > Why were only SD-LoRA and PixArt-Full experiments conducted in the experimental setup, without including SD-Full and PixArt-LoRA related experiments?
>
> In short, this is for fair comparison following the same experimental setup of existing work. Specifically, we found that many works [a,b] used LoRA fine-tuning on SD 1.5. For the sake of fair comparison, we also only applied LoRA fine-tuning for YOSO-SD. As for Pixart, previous works [c] used full fine-tuning, so we also adopted full fine-tuning to achieve the best possible results and ensure the fair comparison.
>
> [a] Trajectory Consistency Distillation: Improved Latent Consistency Distillation by Semi-Linear Consistency Function with Trajectory Mapping
>
> [b] Hyper-SD: Trajectory Segmented Consistency Model for Efficient Image Synthesis, NeurIPS 2024.
>
> [c] PixArt-Σ: Weak-to-Strong Training of Diffusion Transformer for 4K Text-to-Image Generation, ECCV 2024.
>
> > I am curious about how the zero-shot one-step 1024 resolution is effective. As far as I know, 1024 has a different ar_size compared to 512, and the posemb scale is also different. Wouldn't these factors significantly impact the weight distribution of the 1024 model?
>
> This is because the weights of the 1024 model are inherited from the 512 model, and there is a high similarity between their weights. We only merged the compatible model weights. In particular, the overall cosine similarity between the weight of 512 model and the weight of 1024 model is **0.9902**.  Based on this zero-shot adaptation capability to the 1024 model and YOSO's rapid convergence (requiring only around 20k iterations), we hypothesize that the weights learned by YOSO are primarily responsible for generation acceleration which activates some inherent capability within the pre-trained diffusion model. The reason why YOSO-SD-LoRA has some capability to adapt to ControlNet might also be similar. This might also explain why YOSO-SD-LoRA has some compatibility with ControlNet and different unseen customized models finetuned from SD.
>
>
>
>
> > Without retraining, will increasing the sampling steps, similar to LCM, improve the results? If further enhancements are desired for better generated image quality, what would be the subsequent optimization directions based on the proposed approach?
>
> Yes. The YOSO can be used in different sampling steps. We have shown the performance of YOSO in both 1-step and 4-step generation in Table 2. The 4-step generation has a better HPS of 30.50 compared to the 1-step generation which has an HPS of 28.33. This indicates that YOSO can benefit from more sampling steps similar to LCM. To further improve image quality, we may need to develop crafted objectives for few-step generation, or combine with other advanced techniques, such as distribution matching via score distillation [d, e] and human feedback learning [f].
>
> [d] ProlificDreamer: High-Fidelity and Diverse Text-to-3D Generation with Variational Score Distillation, NeurIPS 2023.
>
> [e] Diff-Instruct: A Universal Approach for Transferring Knowledge From Pre-trained Diffusion Models, NeurIPS 2023.
>
> [f] ImageReward: Learning and Evaluating Human Preferences for Text-to-Image Generation, NeurIPS 2023.

---

> > ### Comment · Reviewer_tJNk · 2024-11-26
> >
> > Thank you for your response.
> >
> > - In reference to Tab. 2, could you provide visualized comparison results of YOSO and YOSO-LoRA at different steps (1 & 4) to confirm the performance improvement mentioned in your reply?
> >
> > - Regarding the zero-shot one-step 1024 resolution, in your response, does "YOSO's rapid convergence (requiring only around 20k iterations)" refer to the learning of the merged parameters for the 1024 model? Therefore, does "zero-shot" here not imply no specific training is needed?
> >
> > - The author did not specifically address the concern raised in W2. If possible, could you please provide some clarification on this matter?

---

> > > ### Author Response · Authors · 2024-11-26
> > >
> > > Thank you for your reply. We are glad to provide further clarification to address your concerns.
> > >
> > > > In reference to Tab. 2, could you provide visualized comparison results of YOSO and YOSO-LoRA at different steps (1 & 4) to confirm the performance improvement mentioned in your reply?
> > >
> > > Thanks for your advice. We included a visualization comparison in the Figure 10 located in the appendix. It can be seen that 4-step samples have better visual quality compared to 1-step samples.
> > >
> > > > Regarding the zero-shot one-step 1024 resolution, in your response, does "YOSO's rapid convergence (requiring only around 20k iterations)" refer to the learning of the merged parameters for the 1024 model? Therefore, does "zero-shot" here not imply no specific training is needed?
> > >
> > > The "YOSO's rapid convergence (requiring only around 20k iterations)" refers to the training of YOSO on 512 resolution by initializing from PixArt-$\alpha$-512. The merged model does not need extra training, obtaining by $W_{YOSO-1024} = W_{pixart-1024}-W_{pixart-512}+W_{YOSO-512}$. In particular, "zero-shot" indicates we can train YOSO on 512 resolution by initializing from PixArt-$\alpha$-512, then merge with PixArt-$\alpha$-1024 for one-step generation on 1024 resolution without extra training.
> > >
> > > > The clarity of the narrative in this paper needs improvement; currently, the narrative logic resembles an experimental report. When encountering issues in a new task (unconditional and text-to-image generation / finetuning) or a new model (U-Net or DiT based), a specific strategy is proposed to address the problem. This approach may not be conducive to the coherence expected in a complete paper.
> > >
> > > We clarify that our main focus is text-to-image generation. Unlike existing works [a,b,c] that focus on adapting previous methods [d,e,f] to text-to-image generation, our YOSO is a new model. To validate the effectiveness of our YOSO framework, we fist study a simpler task (i.e., unconditional generation). In addition, we adapt YOSO to text-to-image generation with additional techniques. Our goal is to clearly showcase the fundamental design of our YOSO framework, free from the complexities introduced by additional techniques necessary for text-to-image generation.
> > >
> > > [a] One-step Diffusion with Distribution Matching Distillation, CVPR 2024.
> > >
> > > [b] SwiftBrush: One-Step Text-to-Image Diffusion Model with Variational Score Distillation, CVPR 2024.
> > >
> > > [c] InstaFlow: One Step is Enough for High-Quality Diffusion-Based Text-to-Image Generation, ICLR 2024.
> > >
> > > [d] ProlificDreamer: High-Fidelity and Diverse Text-to-3D Generation with Variational Score Distillation, NeurIPS 2023.
> > >
> > > [e] Diff-Instruct: A Universal Approach for Transferring Knowledge From Pre-trained Diffusion Models, NeurIPS 2023.
> > >
> > > [f] Learning to Generate and Transfer Data with Rectified Flow, ICLR 2023.

---

> > > > ### Comment · Reviewer_tJNk · 2024-11-27
> > > >
> > > > I appreciate the authors for the response. It has addressed most of my concerns and I will raise my rate.

---

### Official Review · Reviewer_gStt · 2024-11-04

**Soundness:** 4
**Presentation:** 3
**Contribution:** 4
**Rating:** 6
**Confidence:** 4

**Summary:**

This paper presents YOSO (You Only Sample Once), a novel approach for high-quality, one-step text-to-image generation that enhances training stability and efficiency by combining elements from diffusion and GAN models. YOSO’s design includes Self-Cooperative Learning, where less noisy “teacher” samples guide more noisy “student” samples to stabilize training, along with a Decoupled Scheduler that applies separate optimization schedules to Adversarial Loss and Consistency Loss, improving sample quality. Additionally, Informative Prior Initialization (IPI) provides a more data-informed starting point for faster convergence and reduced artifacts, while Quick Adaptation facilitates smooth adjustments to v-prediction and zero terminal SNR, essential for stable, one-step sampling. Unlike prior models that require multiple steps, YOSO achieves single-step synthesis with refined adversarial learning, using teacher-student distributions within the model to mitigate instability and mode collapse risks. This efficient and stable framework positions YOSO as a significant advancement for rapid, high-fidelity text-to-image generation.

**Strengths:**

1. The proposed self-cooperative technique to train a one-step sampler seems novel in the diffusion literature, and the author proved it to be very effective. The fact that the less noisy (teacher) samples guide more noisy (student) samples during training is quite interesting.

2. It is interesting to see the connection between the self-cooperative objective and consistency models.

3. The experimental results are quite strong. They outperform other one-step baseline models on CIFAR-10 and also show improvement in the large-scale text-to-image setting.

4. The mathematical analysis is clear and robust, which supports each of its contributions. The derivations for Self-Cooperative Learning and the teacher-student dynamic seems well-structured.

**Weaknesses:**

1. Lack of the ablation study on training techniques. The paper introduces training techniques like Informative Prior Initialization (IPI) and v-prediction, which are valuable training methods that could also benefit other models. However, the lack of an ablation study on these components makes it difficult to assess their specific impact and to directly compare YOSO's performance with other models that could potentially leverage these techniques.

2. Hyperparameter sensitivity. The timestep difference between the teacher and student latents, a key factor in the Decoupled Scheduler, is a hyperparameter that directly impacts training stability and eventual performance. Does it need to be tuned for different datasets? Could you show how robust the method is to this hyperparameter?

[minor comments]
1. Table 2 caption: text-to-to -> text-to
2. Eq (3): remove ‘x’
3. In Eq (3): why did you use different lambda for different t?

**Questions:**

Questions are embedded in the weakness section.

---

> ### Author Response · Authors · 2024-11-25
>
> Thank you for your time and effort in reviewing our paper. We very much appreciate your insightful comments and your recognition of our work.  We hereby address the concerns below.
>
>
> > Lack of the ablation study on training techniques. The paper introduces training techniques like Informative Prior Initialization (IPI) and v-prediction, which are valuable training methods that could also benefit other models. However, the lack of an ablation study on these components makes it difficult to assess their specific impact and to directly compare YOSO's performance with other models that could potentially leverage these techniques.
>
> Thank you for acknowledging our technical contributions. We would like to emphasize that we have conducted comprehensive ablation studies on most of our proposed techniques (including IPI, Latent Perceptual Loss, Decoupled Scheduler, and Annealing Strategy) on text-to-image generation, with results presented in Table 4 of our paper. Here we conducted additional ablation experiments on quick adapt to v-prediction using PixArt-$\alpha$. We evaluate the variants in training PixArt-$\alpha$ on 5k training iterations and 1-step generation. The results are shown below:
> |Method |HPS ↑| AeS ↑|
> |--------|------|-----|
> |$\epsilon$-prediction | 18.03 | 4.93|
> |naive  v-prediction | 22.37 | 5.62|
> |quick adpation to v-prediction | 25.39 | 6.01|
>
> It can be seen that using eps-prediction alone for one-step generation yields inferior performance, while naive v-prediction (directly using v-prediction in training) achieves decent results. However, quick adaptation to v-prediction demonstrates significantly better performance, validating the effectiveness of our proposed quick adaptation to v-prediction.
>
> > Hyperparameter sensitivity. The timestep difference between the teacher and student latents, a key factor in the Decoupled Scheduler, is a hyperparameter that directly impacts training stability and eventual performance. Does it need to be tuned for different datasets? Could you show how robust the method is to this hyperparameter?
>
> Good comment. In all experiments in our paper, we consistently used 250 as the timestep difference between the teacher and student latents in our cooperative adversarial loss. This demonstrates that the hyperparameter is not required to be tuned among different settings. However, we are glad to provide experiments with varying timestep differences. Specifically, we used Lora finetuning SD-v1.5 with 32 batch size and 3000 iterations (note that we used significantly fewer computational resources compared to the YOSO-SD-lora shown in the paper, resulting in lower performance, but this does not affect the relative performance). We report the results below:
>
> | Model | Timestep difference| HPS ↑ | AeS ↑ |
> |--------|------|-----|-----|
> | YOSO | 150 | 25.94 | 5.89 |
> | YOSO | 200 | 26.37 | 6.03 |
> | YOSO | 250 | 26.04 | 5.91 |
> | YOSO | 300 | 25.88 | 5.87 |
>
> The results show that YOSO can achieve consistently good performance by varying the timestep difference.
>
> > In Eq (3): why did you use different lambda for different t?
>
> This is because samples with higher noise have a lower signal-to-noise ratio, making it more difficult to reconstruct samples from them, resulting in higher variance in learning. Therefore, we need a weight that can specify the learning importance of different timesteps. It's worth noting that this approach is actually widely used in diffusion [a] and diffusion distillation [b,c] papers.
>
> [a] Elucidating the Design Space of Diffusion-Based Generative Models, NeurIPS 2022.
>
> [b] Consistency Models, ICML 2023.
>
> [c] Improved Techniques for Training Consistency Models, ICLR 2024.

---

> > ### Comment · Reviewer_gStt · 2024-11-27
> >
> > Thank you for your response. The authors have adequately addressed my concerns. While this work represents a solid contribution to the field, the proposed method and performance do not surprise me as particularly novel or exciting, considering the recent many works that have achieved few-step inference with high-quality images. Anyway, I think this paper meets the standard for acceptance in ICLR, and I maintain my recommendation to borderline accept.

---

> > > ### Author Response · Authors · 2024-11-27
> > >
> > > We thank you for your recognition of our work as a solid contribution to the field. Thanks again for your time and effort in reviewing our work.

---

### Author Response · Authors · 2024-11-25
**General Response**

We sincerely thank the reviewers for their time and thoughtful feedback on our work.

In this work, we introduce YOSO, a novel framework for combining diffusion models and GANs via the proposed self-cooperative adversarial learning. Our work demonstrates strong experimental results and wide application. We appreciate the reviewers' acknowledgment. Specifaclly, "The proposed self-cooperative technique to train a one-step sampler seems novel in the diffusion literature, and the author proved it to be very effective" (Reviewer gStt), "YOSO demonstrates certain advantages across various tasks and metrics, and it is capable of transferring to higher resolutions, image editing, and conditional generation tasks." (Reviewer tJNk), "The visualization results look very good" (Reviewer RjVy), "YOSO’s compatibility with LoRA allows it to produce high-resolution images without the significant computational burden typically associated with GAN-based models" (Reviewer SYJU), YOSO "tackles a relevant application (i.e., ControlNet) with one-step" (Reviewer Awjz).

In what follows, following the reviewers' suggestions and comments, we summarize additional experiments we have conducted to address some common concerns, which will be incorporated to the revision.

1. **Ablation Studies and Training Techniques**:
   As suggested by the reviewers, there is a need for ablation studies to assess the impact of specific training techniques, including Informative Prior Initialization (IPI), v-prediction, the Decoupled Scheduler, and Latent Perceptual Loss. We emphasize that comprehensive ablation studies were conducted and included in the original submission (Table 4) on text-to-image generation. Additionally, we have performed new experiments to further validate the effectiveness of our proposed techniques, such as the quick adaptation to v-prediction and varying timestep differences in the cooperative adversarial loss. These results consistently demonstrate the robustness and performance improvements provided by our methods.

**Table A.** Ablation on timestep difference. The setting is using LoRA for fine-tuning SD 1.5 with 3k iterations.
| Model | Skip | HPS ↑ | AeS ↑ |
|--------|------|-----|-----|
| YOSO | 150 | 25.94 | 5.89 |
| YOSO | 200 | 26.37 | 6.03 |
| YOSO | 250 | 26.04 | 5.91 |
| YOSO | 300 | 25.88 | 5.87 |

**Table B.** Ablation on the model parametrization. The setting is fine-tuning PixArt-$\alpha$ with 5k iterations.
|Method |HPS ↑| AeS ↑|
|--------|------|-----|
|$\epsilon$-prediction | 18.03 | 4.93|
|naive  v-prediction | 22.37 | 5.62|
|quick adpation to v-prediction | 25.39 | 6.01|

2. **Comparison to advanced Diffusion-GAN hybrid Models**
We provide an additional comparison to advanced Diffusion-GAN hybrid Models on CIFAR-10. The results indicate the clear advantage of our proposed YOSO.
| Model | FID↓ |
|-------|--------|
| Diffusion-GAN  | 3.19 |
| Diffusion2GAN  | 3.16 |
| YOSO (Ours) | 1.81 |

3. **Training YOSO from scratch on ImageNet-64**
To further demostrate the ability of YOSO in training from scratch, we conducted additional experiments on ImageNet-64 using the EDM architecture, demonstrating YOSO's effectiveness when trained from scratch.

| Method | NFE | FID (↓) |
|--------|---------------|---------|
| BigGAN-deep  | 1 | 4.06 |
| StyleGAN-XL | 1 | 1.52 |
| ADM  | 250 | 2.07 |
| EDM | 511 | 1.36 |
| CT | 1 | 13.0 |
| iCT  | 1 | 3.25 |
| YOSO （Ours）  | 1 | 2.65 |

4. **Zero-shot FID on COCO**:
   While we argue that zero-shot FID on COCO has limitations as a metric for evaluating modern text-to-image models (Pick-a-Pic [a] found that zero-shot COCO FID can be negatively correlated with human preferences in certain cases), we provided these results for completeness. Our method demonstrates competitive performance compared to existing SOTA models in terms of FID and CLIP Score, further validating its effectiveness.

| Model | Latency ↓ | FID-30k ↓ | CLIP Score-30k ↑ |
|--------|---------|--------|---------|
| **GANs** | | | |
|  StyleGAN-T | 0.10s | 13.90 | - |
|  GigaGAN | 0.13s | 9.09 | - |
| **Oringal Diffusion** | | | |
| SD v1.5 | 2.59s | 13.45 | 0.32 |
| **Diffusion Distillation** | | | |
| LCM (4-step) | 0.26s | 23.62 | 0.30 |
| Diffusion2GAN | 0.09s | 9.29 | - |
| UFOGen | 0.09s | 12.78 | - |
| InstaFlow-0.9B | 0.09s | 13.27 | 0.28 |
| DMD | 0.09s | 14.93 | 0.32 |
| SwiftBrush | 0.09s | 16.67 | 0.29 |
| YOSO (Ours) | 0.09s | 12.35 | 0.31 |
| YOSO（Ours, retrained, full fine-tuned, selected by FID）  | 0.09s | 9.61 | 0.31 |

[a] Pick-a-Pic: An Open Dataset of User Preferences for Text-to-Image Generation, NeurIPS 2023.

We appreciate the reviewers' constructive comments, which have allowed us to improve the clarity and comprehensiveness of our work. We believe that our additional experiments and clarifications address the concerns raised and further demonstrate the robustness, efficiency, and effectiveness of YOSO. Thank you again for your insightful feedback!

---

### Author Response · Authors · 2024-12-03
**Response Summary**

We thank the area chair and all reviewers for your time, insightful suggestions, and valuable comments. Your suggestions have been invaluable in refining our work, and we deeply appreciate the time and effort you dedicated to reviewing our paper. We have carefully addressed all points in our response.

We are encouraged by the reviewers’ positive feedback on various aspects of our work:
* Reviewer gStt: "The proposed self-cooperative technique to train a one-step sampler seems novel in the diffusion literature, and the author proved it to be very effective".
* Reviewer tJNk: "YOSO demonstrates certain advantages across various tasks and metrics, and it is capable of transferring to higher resolutions, image editing, and conditional generation tasks".
* Reviewer RjVy: "The visualization results look very good".
* Reviewer SYJU: "YOSO’s compatibility with LoRA allows it to produce high-resolution images without the significant computational burden typically associated with GAN-based models".
* Reviewer Awjz: YOSO "tackles a relevant application (i.e., ControlNet) with one-step".

We are also pleased that our clarifications and additional experiments have been well-received:
* Reviewer gStt: "I think this paper meets the standard for acceptance in ICLR."
* Reviewer SYJU: "I have no further questions and raised my score."
* Reviewer tJNk: "It has addressed most of my concerns, and I will raise my score."

**Additional Experiments and Improvements**

We also sincerely thank the reviewers for their valuable suggestions, which helped us identify areas for improvement. In response to their feedback, we have made several additional experiments to further strengthen our work:

* **Ablation Studies and Training Techniques** (Reviewer gStt): We performed experiments validating the quick adaptation to v-prediction and varying timestep differences in the cooperative adversarial loss. Results consistently demonstrate the robustness and performance improvements of our methods.
* **Comparison to advanced Diffusion-GAN hybrid Models** (Reviewer RjVy): We provide an additional comparison to advanced Diffusion-GAN hybrid Models on CIFAR-10 and high-resolution dataset FFHQ-1024. The results indicate the clear advantage of our proposed YOSO.
* **Training YOSO from scratch on ImageNet-64** (Reviewer SYJU): To further demonstrate the ability of YOSO in training from scratch, we conducted additional experiments on ImageNet-64. Results show that YOSO achieves comparable performance to state-of-the-art diffusion models and GANs.
* **Super-Resolution (128→512)** (Reviewer RjVy): Experiments demonstrate that YOSO achieves comparable performance to SD upscaler with a ~50x speedup.
* **Zero-shot FID on COCO** (Reviewer Awjz, RjVy): We have provided comparison regarding zero-shot COCO FID. The results show that YOSO achieves competitive performance compared to existing state-of-the-art models. We also observe that YOSO is capable of achieving a lower Zero-shot COCO FID of 8.90 compared to GigaGAN by longer training.
* **Layer Selection for Latent Perceptual Loss (LPL)** (Reviewer Awjz): We follow the reviewers' advice and provide more detailed analysis and experiments, additional ablation studies confirmed that using bottleneck features for LPL consistently outperforms other layer configurations. Moreover, regardless of which layers are used to compute LPL, it is significantly better than MSE.

We believe our additional experiments and clarifications comprehensively address the remaining concerns raised by Reviewer RjVy (regarding additional high-resolution comparison and zero-shot COCO FID) and Reviewer Awjz (regarding layer selection in LPL). Additionally, we explored accelerating SD upscaler using YOSO, addressing the Reviewer RjVy’s interest in this aspect.

We thank all the reviewers again for their time and effort in reviewing our paper and are committed to addressing every issue raised.

Sincerely,

Authors

---

### Meta-Review · Area_Chair_9P3x · 2024-12-21

**Metareview:**

The paper introduces a framework for efficient and high-quality one-step image generation by combining diffusion models with GANs. The authors propose several techniques, including self-cooperative learning, a decoupled scheduler, and informative prior initialization. The paper demonstrates promising results on multiple benchmark datasets.

Following the authors-reviewers discussion, two reviewers increased their ratings. However, other reviewers leaning toward rejection with concerns about: 1) high-resolution generation on certain datasets, 2) inferior FID results, and 3) insufficient ablations to understand the optimal layers/features used for optimization. Subsequently, the authors provided additional responses addressing these three concerns.

Given the promising results, the paper is recommended for acceptance. However, the authors should include the discussions with reviewers in the final version.

**Additional Comments On Reviewer Discussion:**

All reviewers actively participated in the authors-reviewers discussion. The ACs carefully reviewed all responses. While the reviewers raised some concerns, the authors addressed these by providing sufficient experimental results. Consequently, the paper is recommended for acceptance.

---

### Decision · Program_Chairs · 2025-01-22

Accept (Poster)